# DEEP MMD GRADIENT FLOW WITHOUT ADVERSARIAL TRAINING

**Alexandre Galashov**
UCL Gatsby
Google DeepMind
agalashov@google.com

**Valentin De Bortoli**
Google DeepMind
vdebortoli@google.com

**Arthur Gretton**
UCL Gatsby
Google DeepMind
gretton@google.com

## ABSTRACT

We propose a gradient flow procedure for generative modeling by transporting particles from an initial source distribution to a target distribution, where the gradient field on the particles is given by a noise-adaptive Wasserstein Gradient of the Maximum Mean Discrepancy (MMD). The noise-adaptive MMD is trained on data distributions corrupted by increasing levels of noise, obtained via a forward diffusion process, as commonly used in denoising diffusion probabilistic models. The result is a generalization of MMD Gradient Flow, which we call Diffusion-MMD-Gradient Flow or DMMD. The divergence training procedure is related to discriminator training in Generative Adversarial Networks (GAN), but does not require adversarial training. We obtain competitive empirical performance in unconditional image generation on CIFAR10, MNIST, CELEB-A (64 x64) and LSUN Church (64 x 64). Furthermore, we demonstrate the validity of the approach when MMD is replaced by a lower bound on the KL divergence.

## 1 INTRODUCTION

In recent years, generative models have achieved impressive capabilities on image Saharia et al. (2022), audio Le et al. (2023) and video generation Ho et al. (2022) tasks but also protein modeling Watson et al. (2023) and 3d generation Poole et al. (2023). Diffusion models (Sohl-Dickstein et al., 2015; Ho et al., 2020; Song et al., 2021; Rombach et al., 2022) underpin these new methods. In these models, we learn a backward denoising diffusion process via denoising score matching (Hyvärinen, 2005; Vincent, 2011). This backward process corresponds to the time-reversal of a forward noising process. At sampling time, starting from random Gaussian noise, diffusion models produce samples by discretizing the backward process.

One challenge that arises when applying these models in practice is that the Stein score (that is, the gradient log of the current noisy density) becomes ill-behaved near the data distribution (Yang et al., 2024): the diffusion process needs to be slowed down at this point, which incurs a large number of sampling steps near the data distribution. Indeed, if the manifold hypothesis holds Tenenbaum et al. (2000); Fefferman et al. (2016); Brown et al. (2022) and the data is supported on a lower dimensional space, it is expected that the score will explode for noise levels close to zero, to ensure that the backward process concentrates on this lower dimensional manifold Bortoli (2022); Pidstrigach (2022); Chen et al. (2023). While strategies exist to mitigate these issues, they trade-off the quality of the output against inference speed, see for instance (Song et al., 2023; Xu et al., 2024; Sauer et al., 2023).

Generative Adversarial Networks (GANs) (Goodfellow et al., 2014) represent an alternative popular generative modelling framework (Brock et al., 2019; Karras et al., 2020a). Candidate samples are produced by a *generator*: a neural net mapping low dimensional noise to high dimensional images. The generator is trained in alternation with a *discriminator*, which is a measure of discrepancy between the generator and target images. An advantage of GANs is that image generation is fast once the GAN is trained (Xiao et al., 2022), although image samples are of lower quality than for the best diffusion models (Ho et al., 2020; Rombach et al., 2022). When learning a GAN model, the main challenge arises due to the presence of the generator, which must be trained adversarially alongside the discriminator. This requires careful hyperparameter tuning (Brock et al., 2019; Karras

et al., 2020b; Liu et al., 2021), without which GANs may suffer from training instability and mode collapse (Arora et al., 2017; Kodali et al., 2017; Salimans et al., 2016).

Nonetheless, the process of GAN design has given rise to a strong understanding of discriminator functions, and a wide variety of different divergence measures have been applied. These fall broadly into two categories: the integral probability metrics (among which, the Wasserstein distance (Arjovsky et al., 2017; Gulrajani et al., 2017; Genevay et al., 2018) and the Maximum Mean Discrepancy (Li et al., 2017; Binkowski et al., 2018; Arbel et al., 2018)) and the f-divergences (Goodfellow et al., 2014; Nowozin et al., 2016; Mescheder et al., 2018; Brock et al., 2019). While it would appear that f-divergences ought to suffer from the same shortcomings as diffusions when the target distribution is supported on a submanifold Arjovsky et al. (2017), the divergences used in GANs are in practice variational lower bounds on their corresponding f-divergences (Nowozin et al., 2016), and in fact behave closer to IPMs in that they do not require overlapping support of the target and generator samples, and can metrize weak convergence (Arbel et al., 2021, Proposition 14) and (Zhang et al., 2018) (there remain important differences, however: notably, f-divergences and their variational lower bounds need not be symmetric in their arguments).

A natural question then arises: is it possible to define a Wasserstein gradient flow (Ambrosio et al., 2008; Santambrogio, 2015) using a GAN discriminator as a divergence measure? In this setting, the divergence (discriminator) provides a gradient field directly onto a set of particles (rather than to a generator), transporting them to the target distribution. Contributions in this direction include the MMD flow Arbel et al. (2019); Hertrich et al. (2024), which defines a Wasserstein Gradient Flow on the Maximum Mean Discrepancy (Gretton et al., 2012); and the KALE (KL approximate lower-bound estimator) flow Glaser et al. (2021), which defines a Wasserstein gradient flow on a KL lower bound of the kind used as a GAN discriminator based on an f-divergence (Nowozin et al., 2016). We describe the MMD and its corresponding Wasserstein gradient flow in Section 2. These approaches employ fixed function classes (namely, reproducing kernel Hilbert spaces) for the divergence, and are thus not suited to high dimensional settings such as images. Moreover, we show in this work that even for simple examples in low dimensions, an adaptive discriminator ensures faster convergence of a source distribution to the target, see Section 3.

A number of more recent approaches employ trained neural net features in divergences for a subsequent gradient flow (e.g. Fan et al., 2022; Franceschi et al., 2023). Broadly speaking, these works used adversarial means to train a *series* of discriminator functions, which are then applied in sequence to a population of particles. While more successful on images than kernel divergences, the approaches retain two shortcomings: they still require adversarial training (on their own prior output), with all the challenges that this entails; and their empirical performance falls short in comparison with modern diffusions and GANs (see related work in Section 6 for details).

In the present work, we propose a novel Wasserstein Gradient flow on a noise-adaptive MMD divergence measure, leveraging insights from both GANs and diffusion models. To *train the discriminator*, we start with clean data, and use a forward diffusion process from (Ho et al., 2020) to produce noisy versions of the data with given levels of noise (data with high levels of noise are analogous to the output of a poorly trained generator, whereas low noise is analogous to a well trained generator). The added noise is always Gaussian. For a given level of noise, we train a noise conditional MMD discriminator to distinguish between the clean and the noisy data, using a single network across all noise levels. This allows us to have better control over the discriminator training procedure than would be achievable with a GAN generator at different levels of refinement, where this control is implicit and hard to characterize.

To *draw new samples*, we propose a novel noise-adaptive version of MMD gradient flow (Arbel et al., 2019). Starting from Gaussian distribution, we move them in the direction of the target distribution by following MMD Gradient flow (Arbel et al., 2019), adapting our MMD discriminator to the corresponding level of noise. See Section 4 for details. This allows us to have a fine grained control over the sampling process. As a final challenge, MMD gradient flows have previously required large populations of interacting particles for the generation of novel samples, which is expensive (quadratic in the number of particles) and impractical. In Section 5, we propose a scalable approximate sampling procedure for a case of a linear base kernel, which allows *single* samples to be generated with a very little loss in quality, at cost independent of the number of particles used in training. The MMD is an instance of an integral probability metric, however many GANs have been designed using discriminators derived from f-divergences. Section E demonstrates how our approach can be applied

to such divergences, using a lower bound on the KL divergence as an illustration. Section 6 contains a review of alternative approaches to using GAN discriminators for sample generation. Finally, in Section 7, we show that our method, Diffusion-MMD-gradient flow (DMMD), yields competitive performance in generative modeling on 2-D datasets as well as in unconditional image generation on CIFAR10 (Krizhevsky & Hinton, 2009), MNIST, CELEB-A, LSUN Church.

## 2 BACKGROUND

In this section, we define the MMD as a GAN discriminator, then describe Wasserstein gradient flow as it applies for this divergence measure.

**MMD GAN.** Let $\mathcal{X} \subset \mathbb{R}^D$ and $\mathcal{P}(\mathcal{X})$ be the set of probability distributions on $\mathcal{X}$. Let $P \in \mathcal{P}(\mathcal{X})$ be the *target* (data) distribution and $Q_\psi \in \mathcal{P}(\mathcal{X})$ be a distribution associated with a *generator* parameterized by $\psi \in \mathbb{R}^L$. Let $\mathcal{H}$ be Reproducing Kernel Hilbert Space (RKHS), see (Schölkopf & Smola, 2018) for details, for some kernel $k : \mathcal{X} \times \mathcal{X} \to \mathbb{R}$. The Maximum Mean Discrepancy (MMD) (Gretton et al., 2012) between $Q_\psi$ and $P$ is defined as $\mathrm{MMD}(Q_\psi, P) = \sup_{\|f\|_\mathcal{H} \leq 1}\{\mathbb{E}_{Q_\psi}[f(X)] - \mathbb{E}_P[f(X)]\}$. We refer to the function $f_{Q_\psi,P}$ that attains the supremum as the *witness function*,

$$f_{Q_\psi,P}(z) \propto \int k(x,z)\mathrm{d}Q_\psi(x) - \int k(y,z)\mathrm{d}P(y), \tag{1}$$

which will be essential in defining our gradient flow. Given $X^N = \{x_i\}_{i=1}^N \sim Q_\psi^{\otimes N}$ and $Y^M = \{y_i\}_{i=1}^M \sim P^{\otimes M}$, the empirical witness function is known in closed form, $\hat{f}_{Q_\psi,P}(x) \propto \frac{1}{N}\sum_{i=1}^N k(x_i,x) - \frac{1}{M}\sum_{j=1}^M k(y_j,x)$, and an unbiased estimate of $\mathrm{MMD}^2$ (Gretton et al., 2012) is likewise straightforward. In the MMD GAN (Binkowski et al., 2018; Li et al., 2017), the kernel is

$$k(x,y) = k_{\mathrm{base}}(\phi(x;\theta), \phi(y;\theta)), \tag{2}$$

where $k_{\mathrm{base}}$ is a base kernel and $\phi(\cdot;\theta) : \mathcal{X} \to \mathbb{R}^K$ are neural networks *discriminator* features with parameters $\theta \in \mathbb{R}^H$. We use the modified notation $\mathrm{MMD}_u^2[X^N, Y^M; \theta]$ to highlight the functional dependence on the discriminator parameters. The MMD is an Integral Probability Metric (IPM) (Muller, 1997), and thus well defined on distributions with disjoint support: this argument was made in favor of IPMs by Arjovsky et al. (2017). Note further that the Wasserstein GAN discriminators of Arjovsky et al. (2017); Gulrajani et al. (2017) can be understood in the MMD framework, when the base kernel is linear. Indeed, it was observed by Genevay et al. (2018) that requiring closer approximation to a true Wasserstein distance resulted in decreased performance in GAN image generation, likely due to the the exponential dependence of sample complexity on dimension for the exact computation of the Wasserstein distance; this motivates an interpretation of these discriminators simply as IPMs using a class of linear functions of learned features. We further note that the variational lower bounds used in approximating f-divergences for GANs share with IPMs the property of being well defined on distribtions with disjoint support Nowozin et al. (2016); Arbel et al. (2021), although they need not be symmetric in their arguments. Finally, while $Q_\psi$ and $\theta$ are trained adversarially in GANs, our setting will only require us to learn the discriminator parameter $\theta$.

**Wasserstein gradient flows.** Instead of a GAN generator, we can move a sample of particles along the Wasserstein Gradient flow associated with the discriminator (Ambrosio et al., 2008). Let $\mathcal{P}_2(\mathcal{X})$ be a set of probability distributions on $\mathcal{X}$ with a finite second moment equipped with the 2-Wasserstein distance. Let $\mathcal{F}(\nu) : \mathcal{P}_2(\mathcal{X}) \to \mathbb{R}$ be a functional defined over $\mathcal{P}_2(\mathcal{X})$ with a property that $\arg\inf_\nu \mathcal{F}(\nu) = P$. We consider the problem of transporting mass from an initial distribution $\nu_0 = Q$ to a target distribution $\mu = P$, finding a continuous path $(\nu_t)_{t \geq 0}$ starting from $\nu_0$ that converges to $\mu$. This problem is studied in Optimal Transport theory (Villani, 2008; Santambrogio, 2015). This path can be discretized as a sequence of random variables $(X_n)_{n \in \mathbb{N}}$ such that $X_n \sim \nu_n$,

$$X_{n+1} = X_n - \gamma\nabla\mathcal{F}'(\nu_n)(X_n), \quad X_0 \sim Q, \tag{3}$$

where $\eta > 0$ and $\mathcal{F}'(\nu_n)(X_n)$ is the first variation of $\mathcal{F}$ associated with the Wasserstein gradient, see (Ambrosio et al., 2008; Arbel et al., 2019) for precise definitions. As $n \to \infty$ and $\gamma \to 0$, depending on the conditions on $\mathcal{F}$, the process (3) will convergence to the gradient flow as a continuous time limit (Ambrosio et al., 2008).

**MMD gradient flow.** For a choice $\mathcal{F}(\nu) = \mathrm{MMD}^2[\nu, P]$ and a fixed kernel, conditions for convergence of the process in (3) to $P$ are given by Arbel et al. (2019). Moreover, the first variation of $\mathcal{F}'(\nu) = f_{\nu,P} \in \mathcal{H}$ is the witness function defined earlier.[1] Using (1)-(3), the discretized MMD gradient flow for any $n \in \mathbb{N}$ is given by

$$X_{n+1} = X_n - \gamma \nabla f_{\nu_n, P}(X_n), \qquad X_0 \sim Q. \tag{4}$$

This provides an algorithm to (approximately) sample from the target distribution $P$. We remark that Arbel et al. (2019); Hertrich et al. (2024) used a kernel with fixed hyperparameters. In the next section, we will argue that even for RBF kernels (where only the bandwidth is chosen), faster convergence will be attained using kernels that adapt during the gradient flow. Details of kernel choice for alternative approaches are given in related work (Section 6).

## 3 A MOTIVATION FOR ADAPTIVE KERNELS

In this section, we demonstrate the benefit of using an *adaptive* kernel when performing MMD gradient flow. We show that even in the simple setting of Gaussian sources and targets, an adaptive kernel improves the convergence of the flow. Let $k_\alpha(x, y) = \alpha^{-d} \exp[-\|x - y\|^2/(2\alpha^2)]$ be the normalized Gaussian kernel. For any $\mu \in \mathbb{R}^d$ and $\sigma > 0$ we denote by $\pi_{\mu,\sigma}$ the Gaussian distribution with mean $\mu$ and covariance matrix $\sigma^2 \mathrm{Id}$. We denote $\mathrm{MMD}_\alpha$ the MMD associated with $k_\alpha$.

**Proposition 3.1.** *For any $\mu_0 \in \mathbb{R}^d$ and $\sigma > 0$, let $\alpha^\star$ be given by*

$$\alpha^\star = \mathrm{argmax}_{\alpha \geq 0} \|\nabla_{\mu_0} \mathrm{MMD}_\alpha^2(\pi_{0,\sigma}, \pi_{\mu_0,\sigma})\|.$$

*Then, we have that*

$$\alpha^\star = \mathrm{ReLU}(\|\mu_0\|^2/(d+2) - 2\sigma^2)^{1/2}. \tag{5}$$

The result is proved in Appendix K. The quantity $\|\nabla_{\mu_0} \mathrm{MMD}_\alpha^2(\pi_{0,\sigma}, \pi_{\mu_0,\sigma})\|$ represents how much the mean of the Gaussian $\pi_{\mu_0,\sigma}$ is displaced by a flow w.r.t. $\mathrm{MMD}_\alpha^2$. We want $\|\nabla_{\mu_0} \mathrm{MMD}_\alpha^2(\pi_{0,\sigma}, \pi_{\mu_0,\sigma})\|$ as large as possible as it denotes the *maximum displacement possible*.

We show that $\alpha^\star$ maximizing this displacement is given by (5). Assuming $\sigma > 0$ is fixed, it is notable that this quantity depends on $\|\mu_0\|$, i.e. the distance between the two distributions. This observation justifies our approach of following an *adaptive* MMD flow at inference time. We further highlight the phase transition behaviour of Proposition 3.1: once the Gaussians are sufficiently close, the optimal kernel width is zero (note that this phase transition would not be observed in the simpler Dirac GAN example of Mescheder et al. (2018), where the source and target distributions are Dirac masses with no variance). This phase transition suggests that the flow associated with MMD benefits *less* from adaptivity as the supports of the distributions overlap. We exploit this observation by introducing an optional denoising stage to our procedure; see the end of Section 4.

In practice, it is not desirable to approximate the distributions of interest by Gaussians, and richer neural network kernel features $\phi(x; \theta)$ are used (see Section 7). Arbel et al. (2018) describe approaches to optimize the MMD parameters for GAN training, which serve as proxies for convergence speed: it is not sufficient simply to maximize the MMD, since the witness function should remain Lipschitz to ensure convergence (Arbel et al., 2018, Proposition 2). Regularization of the witness is achieved in practice by controlling the gradient of the witness function; we take a similar approach in Section 4.

## 4 DIFFUSION MAXIMUM MEAN DISCREPANCY GRADIENT FLOW

In this section, we present *Diffusion Maximum Mean Discrepancy gradient flow* (DMMD), a new generative model with a training procedure of MMD discriminator which does not rely on adversarial training, and leverages ideas from diffusion models. The sampling part of DMMD consists in following a noise adaptive variant of MMD gradient flow.

---

[1] In the case of variational lower bounds on f-divergences, the witness function is still well defined, and the first variation takes the same form in respect of this witness function: see Glaser et al. (2021) for the case of the KL divergence.

**Adversarial-free training of noise conditional discriminators.** In order to train a discriminator without adversarial training, we propose to use insights from GANs training. In a GAN setting, at the beginning of the training, the generator is randomly initialized and therefore produces samples close to random noise. This would produce a coarse discriminator since it is trained to distinguish clean data from random noise. As the training progresses and the generator improves, so too does the discriminative power of the discriminator. This discriminator behavior is central in the training of GANs (Goodfellow et al., 2014). We propose a way to replicate this gradually improving behavior without adversarial training and instead relying on principles from diffusion models (Ho et al., 2020).

The forward process in diffusion models allows us to generate a probability path $P_t, t \in [0, 1]$, such that $P_0 = P$, where $P$ is our target distribution and $P_1 = \mathrm{N}(0, \mathrm{Id})$ is a Gaussian noise. Given samples $x_0 \sim P_0 = P$, the samples $x_t | x_0$ are given by

$$x_t = \alpha_t x_0 + \beta_t \epsilon, \quad \epsilon \in \mathrm{N}(0, \mathrm{Id}), \tag{6}$$

with $\alpha_0 = \beta_1 = 1$ and $\alpha_1 = \beta_0 = 0^2$. From the form of the $x_t | x_0$, we observe that for low noise level $t$, the samples $x_t$ are very close to the original data $x_0$, whereas for the large values of $x_t$ they are close to a unit Gaussian random variable. Using the GANs terminology, $x_t$ could be thought as the output of a generator such that for high/low noise level $t$, it would correspond to *undertrained* / *well-trained* generator. Using this insight, for each noise level $t \in [0, 1]$, we define a discriminator $\mathrm{MMD}^2(P_t, P; t, \theta)$ using the kernel of type (2) with noise-conditional discriminator features $\phi(x; t; \theta)$ parameterized by a Neural Network with learned parameters $\theta$. We consider the following noise-conditional loss function

$$\mathcal{L}(\theta, t) = -\mathrm{MMD}^2(P_t, P; t, \theta) \tag{7}$$

where the minus sign comes from the fact that our aim is to maximize the squared MMD. In addition, we regularize this loss with $\ell_2$-penalty (Binkowski et al., 2018) denoted $\mathcal{L}_{\ell_2}(\theta, t)$ as well as with the gradient penalty (Binkowski et al., 2018; Gulrajani et al., 2017) denoted $\mathcal{L}_\nabla(\theta, t)$, see Appendix C.2 for the precise definition of these two losses. The total noise-conditional loss is then given as

$$\mathcal{L}_{\mathrm{tot}}(\theta, t) = \mathcal{L}(\theta, t) + \lambda_{\ell_2} \mathcal{L}_{\ell_2}(\theta, t) + \lambda_\nabla \mathcal{L}_\nabla(\theta, t), \tag{8}$$

for a suitable choice of hyperparameters $\lambda_{\ell_2} \geq 0, \lambda_\nabla \geq 0$. Finally, the total loss is given as $\mathcal{L}_{\mathrm{tot}}(\theta) = \mathbb{E}_{t \sim U[0,1]} [\mathcal{L}_{\mathrm{tot}}(\theta, t)]$, where $U[0, 1]$ is a uniform distribution. In practice, we use sampled-based unbiased estimator of MMD, see Appendix C.2. The procedure is described in Algorithm 1.

**Adaptive gradient flow sampling.** In order to produce samples from $P$, we use the adaptive MMD gradient flow with noise conditional discriminators $\mathrm{MMD}^2[P_t, P; t; \theta^\star]$, where $\theta^\star$ are the discriminator parameters obtained using Algorithm 1. Let $t_i = t_{\min} + i\Delta t, i = 0, \dots, T$ be the noise discretisation, where $\Delta t = (t_{\max} - t_{\min})/T$ such that $t_0 = t_{\min}, t_T = t_{\max}$ for some $t_{\min} = \epsilon$ and $t_{\max} = 1 - \epsilon$, where $\epsilon \ll 1$. We sample $N_{\mathrm{p}}$ initial particles $\{Z^i | Z^i \sim \mathrm{N}(0, \mathrm{Id})\}_{i=1}^{N_{\mathrm{p}}}$. For each $t$, we follow MMD gradient flow (4) for $N_{\mathrm{s}}$ steps with learning rate $\eta > 0$

$$Z_t^{i,n+1} = Z_t^{i,n} - \eta \nabla f_{\nu_{N_{\mathrm{p}},n}^t, P}(Z_t^{i,n}, t; \theta^\star). \tag{9}$$

Here $\nu_{N_{\mathrm{p}},n}^t = 1/N_{\mathrm{p}} \sum_{i=1}^{N_{\mathrm{p}}} \delta_{Z_{i,n}^t}$ is the empirical distribution of particles $\{Z_t^{i,n}\}_{i=1}^{N_{\mathrm{p}}}$ at the noise level $t$ and the iteration $n$, $\delta$ is a Dirac mass measure. The function $f_{\nu_{N_{\mathrm{p}},n}^t, P}(z, t; \theta^\star)$ is adapted from equation (1) where $\nu$ is replaced by this empirical distribution. After following the gradient flow (9) for $N_{\mathrm{s}}$ steps, we initialize a new gradient flow with initial particles $Z_{t-\Delta t}^{i,0} = Z_t^{i,N_{\mathrm{s}}}$ for each $i = 1, \dots, N_{\mathrm{p}}$, with the decreased level of noise $t - \Delta t$. The recurrence is initialized with $Z_{t_{\max}}^{i,0} = Z^i$ where $\{Z^i\}_{i=1}^{N_{\mathrm{p}}}$ are the initial particles. This procedure corresponds to running $T + 1$ consecutive MMD gradient flows for $N_{\mathrm{s}}$ iterations each, gradually decreasing the noise level $t$ from $t_{\max}$ to $t_{\min}$. The resulting particles $\{Z_{t_{\min}}^{i,N_{\mathrm{s}}}\}_{i=1}^{N_{\mathrm{p}}}$ are used as samples from $P$. See Algorithm 2.

In practice, we sample (once) a large batch $N_c$ of $\{X_0^j\}_{j=1}^{N_c} \sim P^{\otimes N_c}$ from the data distribution and denote by $\hat{P}_{N_c}(X_0)$ the corresponding empirical distribution. Then we use the empirical witness

---

$^2$Different schedules $(\alpha_t, \beta_t)$ are available in the literature. We focus on Variance Preserving SDE ones Song et al. (2021) here

function $f_{\nu^t_{N_p,n}, \hat{P}_{N_c}(X_0)}(z, t; \theta^\star)$ given by

$$\frac{1}{N_p} \sum_{i=1}^{N_p} k_{\text{base}}(\phi(Z_t^{n,i}, t; \theta^\star), \phi(z, t; \theta^\star)) - \frac{1}{N_c} \sum_{j=1}^{N_c} k_{\text{base}}(\phi(X_0^j, t; \theta^\star), \phi(z, t; \theta^\star)). \tag{10}$$

---

**Algorithm 1** Train noise-conditional MMD discriminator

**Input:** Dataset $\mathcal{D} = \{x_i\}_{i=1}^N$
Discriminator features $\phi(x, t; \theta)$ with parameters $\theta \in \mathbb{R}^K$
$\lambda_\nabla \geq 0, \lambda_{\ell_2} \geq 0$ - gradient and $\ell_2$ penalty coefficients
$\gamma > 0$ – learning rate
$N_{\text{iter}}$ – number of iterations, $B$ – batch size
$N_{\text{noise}}$ – number of noise levels per batch
**for** $i = 1$ **to** $N_{\text{iter}}$ **do**
  Sample a batch $B$ of clean particles
  $X_0 \sim P(X_0)$
  **for** $n = 1$ **to** $N_{\text{noise}}$ **do**
    Sample noise level $t_n \sim U[0, 1]$
    Sample $X_{t_n} \sim p(X_{t_n}|X_0, t_n)$
    Let the clean and noisy features be
    $\phi_{t_n}^{X_0} = \phi(X_0, t_n; \theta)$
    $\phi_{t_n}^{X_{t_n}} = \phi(X_{t_n}, t_n; \theta)$
    For linear base kernel (11), use optimized (19) to compute MMD loss (7)
    Compute the loss $\mathcal{L}_{\text{tot}}(\theta, t_n)$ using (8)
  **end for**
  Compute total loss
  $\mathcal{L}_{\text{tot}}(\theta) = \frac{1}{N_{\text{noise}}} \sum_{n=1}^{N_{\text{noise}}} \mathcal{L}_{\text{tot}}(\theta, t_n)$
  Update discriminator features
  $\theta \leftarrow \text{ADAM}(\theta, \mathcal{L}_{\text{tot}}(\theta), \gamma)$
**end for**

---

**Algorithm 2** Noise-adaptive MMD gradient flow

**Inputs:** $T$ – number of noise levels
$t_{\max}, t_{\min}$ – maximum/minimum noise levels
$N_s$ – number of gradient flow steps per noise level
$\eta > 0$ – gradient flow learning rate
$N_p$ – number of noisy particles
Batch of clean particles $X_0 \sim \mathcal{P}_0$.
**Steps:** Sample initial particles $Z \sim \text{N}(0, \text{Id})$
Set $\Delta t = (t_{\max} - t_{\min})/T$
**for** $i = T$ **to** $0$ **do**
  Set the noise level $t = t_{\min} + i\Delta t$
  Set $Z_t^0 = Z$
  **for** $n = 0$ **to** $N_s - 1$ **do**
    Use   (10)   to   compute $f_{\nu^t_{N_p,n}, \hat{P}_{N_c}(X_0)}(Z_t^n, t; \theta^\star)$
    $Z_t^{n+1} = Z_t^n - \eta \nabla f_{\nu^t_{N_p,n}, \hat{P}_{N_c}(X_0)}(Z_t^n, t; \theta^\star)$
  **end for**
  Set $Z = Z_t^N$
**end for**
Output $Z$

---

**Final denoising.** In diffusion models (Ho et al., 2020), it is common to use a denoising step at the end to improve sample quality. We found empirically that a few MMD gradient flow steps at the end of the sampling with a higher learning rate $\eta$ allowed to reduce noise and improve performance.

## 5   SCALABLE DMMD WITH LINEAR KERNEL

The computational complexity of the MMD estimate on two sets of $N$ samples is $O(N^2)$. This is likewise the cost of computing the witness function (10) at $N$ particles, when computed using $N$ clean and noisy particles. However, using linear base kernel (see (2))

$$k_{\text{base}}(x, y) = \langle x, y \rangle, \tag{11}$$

allows to reduce the computation complexity of both quantities down to $O(N)$, see Appendix C.3. We consider the average noise conditional discriminator features on the *whole* dataset

$$\bar{\phi}(X_0, t; \theta^\star) = \frac{1}{N} \sum_{i=1}^N \phi(X_0^i, t; \theta^\star). \tag{12}$$

With the linear kernel (11) we can use average features (12) in the second term of (10). In practice, we can precompute these $K$-dimensional features for $T$ timesteps and store them in memory for later use for sampling purposes, with a storage cost of $O(TK)$. This removes the need to store store the data sample $\{X_0^j\}_{j=1}^N$, since we only retain the *average feature vector* of this sample.

**Approximate sampling procedure.** MMD gradient flow (9) requires us to use multiple interacting particles $Z$ to produce samples, where the interaction is captured by the first term in (10). In practice this means that the performance will depend on the number of these particles. In this

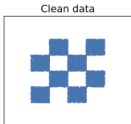 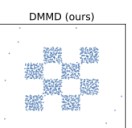 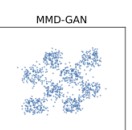 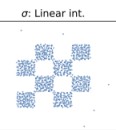 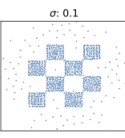 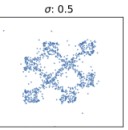 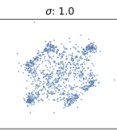

Clean data  DMMD (ours)  MMD-GAN  $\sigma$: Linear int.  $\sigma$: 0.1  $\sigma$: 0.5  $\sigma$: 1.0

Figure 1: Samples from MMD Gradient flow with different parameters for the RBF kernel.

section, we propose an approximation to MMD gradient flow with a linear base kernel (11) which allows us to sample particles *individually and independently*, therefore removing the need for multiple particles. For a linear kernel, the interaction term in (10) for a particle $Z$, equals to $\langle \frac{1}{N_p} \sum_{i=1}^{N_p} \phi(Z_t^{n,i}, t; \theta^\star), \phi(Z, t; \theta^\star) \rangle$. For a large number of particles $N_p$, the contribution of each particle $Z_{n,i}^t$ on the interaction term with $Z$ will be small. For a sufficiently large $N_p$, we hypothesize that $\frac{1}{N_p} \sum_{i=1}^{N_p} \phi(Z_t^{n,i}, t; \theta^\star) \approx \frac{1}{N} \sum_{j=1}^{N} \phi(X_t^j, t; \theta^\star)$, where $N$ is the size of the dataset and $X_t^j$ are produced by the forward diffusion process (6) applied to each $X_0^j$. Thus, we consider an approximate witness function

$$\hat{f}_{P_t, P}(z) = \langle \phi(z, t; \theta^\star), \bar{\phi}(X_t, t; \theta^\star) - \bar{\phi}(X_0, t; \theta^\star) \rangle, \tag{13}$$

with $\bar{\phi}(X_t, t; \theta^\star)$ precomputed using (12). Once again, crucially, we have no need to store the sample $\{X_t^j\}_{j=1}^N$, since we need only retain its feature mean. We may then sample a *single* particle $Z \sim N(0, \mathrm{Id})$ and follow noise-adaptive MMD gradient flow with (13), i.e. $Z_t^{n+1} = Z_t^n - \eta \nabla \hat{f}_{P_t, P}(Z_t^n)$. The corresponding algorithm is described in Appendix C.4.

## 6 RELATED WORK

**Adversarial training and MMD-GAN.** Integral Probability Metrics (IPMs) are good candidates to define discriminators in the context of generative modeling, since they are well defined even in the case of distributions with non-overlapping support (Muller, 1997). Moreover, implementations of f-divergence discriminators in GANs rely on variational lower bounds (Nowozin et al., 2016): as noted earlier, these share useful properties of IPMs in theory and in practice (notably, they remain well defined for distributions with disjoint support, and may metrize weak convergence for sufficiently rich witness function classes (Arbel et al., 2021, Proposition 14) and (Zhang et al., 2018)). Several works (Arjovsky et al., 2017; Gulrajani et al., 2017; Genevay et al., 2018; Li et al., 2017; Binkowski et al., 2018) have exploited IPMs as discriminators for training GANs, where the IPMs are MMDs using (linear or nonlinear) kernels defined on learned neural net features, making them suited to high dimensional settings such as image generation. Interpreting the IPM-based GAN discriminator as a squared MMD yields an interesting theoretical insight: Franceschi et al. (2022) show that training a GAN with an IPM objective implicitly optimizes $\mathrm{MMD}^2$ in the Neural Tangent Kernel (NTK) limit (Jacot et al., 2018). IPM GAN discriminators are trained jointly with the generator in a min-max game. Adversarial training is challenging, and can suffer from instability, mode collapse, and misconvergence (Xiao et al., 2022; Binkowski et al., 2018; Li et al., 2017; Arora et al., 2017; Kodali et al., 2017; Salimans et al., 2016). Note that once a GAN has been trained, the samples can be refined via MCMC sampling in the generator latent space (e.g., using kinetic Langevin dynamics; see Ansari et al., 2021; Che et al., 2020; Arbel et al., 2021).

**Discriminator flows for generative modeling.** Wasserstein Gradient flows (Ambrosio et al., 2008; Santambrogio, 2015) applied to a GAN discriminator are informally called *discriminator flows*, see (Franceschi et al., 2023). A number of recent works have focused on replacing a GAN generator by a discriminator flow. Fan et al. (2022) propose a discretisation of JKO (Jordan et al., 1998) scheme to define a Kullback-Leibler (KL) divergence gradient flow. Other approaches have used a discretized interactive particle-based approach instead of JKO, similar to (3). Heng et al. (2023); Franceschi et al. (2023) build such a flow based on f-divergences, whereas Franceschi et al. (2023) focuses on MMD gradient flow. In all these works, an explicit generator is replaced by a corresponding discriminator flow. The sampling process during training is as follows: Let $Y_k$ be the samples produced at training iteration $k$ by the gradient flow $\mathcal{F}_k$ induced by the discriminator $\mathcal{D}_k$ applied to samples $Y_{k-1}$ from the previous iteration. We denote this by $Y_k \leftarrow \mathcal{F}_k(\mathcal{D}_k, Y_{k-1})$. Then, the discriminator at iteration $k+1$ is trained on samples $Y_k$. A challenge of this process is that the training sample for the next discriminator will be determined by the previous discriminators, and thus the generation process is

still adversarial: particle transport minimizes the previous discriminator value, and the subsequent discriminator is maximized on these particles. Consequently, it is difficult to control or predict the overall sample trajectory from the initial distribution to the target, which might explain the performance shortfall of these methods in image generation settings. By contrast, we have explicit control over the training particle trajectory via the forward noising diffusion process.

Furthermore, these approaches (except for Heng et al., 2023) require to store all intermediate discriminators $\mathcal{D}_1, \ldots, \mathcal{D}_N$ throughout training ($N$ is the total number of training iterations). These discriminators are then used to produce new samples by applying the sequence of gradient flows $\mathcal{F}_N(\mathcal{D}_N, \cdot) \circ \ldots \circ \mathcal{F}_1(\mathcal{D}_1, \cdot)$ to $Y_0$ sampled from the initial distribution. This creates a large memory overhead. An alternative is to use pretrained features obtained elsewhere or a fixed kernel with empirically selected hyperparameters (see Hertrich et al., 2024; Hagemann et al., 2024; Altekrüger et al., 2023), however this limits the applicability of the method. To the best of our knowledge, our approach is the first to demonstrate the possibility to train a discriminator without adversarial training, such that this discriminator can then be used to produce samples with a gradient flow. Unlike the alternatives, our approach does not require to store intermediate discriminators.

MMD **for diffusion refinement/regularization.** MMD has been used to either regularize training of diffusion models (Li & van der Schaar, 2024) or to finetune them (Aiello et al., 2024) for fast sampling. The MMD kernel in these works has the form (2) with Inception features (Szegedy et al., 2015). Our method removes the need to use pretrained features by training the MMD discriminator.

**Diffusion models.** Diffusion models (Sohl-Dickstein et al., 2015; Ho et al., 2020; Song et al., 2021) represent a powerful new family of generative models due to their strong empirical performance in many domains (Saharia et al., 2022; Le et al., 2023; Ho et al., 2022; Watson et al., 2023; Poole et al., 2023). Unlike GANs, diffusion models do not require adversarial training. At training time, a denoiser is learned for multiple noise levels. As noted above, our work borrows from the training of diffusion models, as we train a discriminator on multiple noise levels of the forward diffusion process (Ho et al., 2020). This gives better control of the training samples for the (noise adapted) discriminator than using an incompletely trained GAN generator. We may also consider the setting of flow matching, which is related to the diffusion setting. The potential advantages of our approach to flow matching are discussed in Appendix G.

**Predictor-corrector.** Backwards diffusion might produce samples at time $t$ which do not correspond to the forward process at the same time. To fix this discrepancy, one can leverage the predictor-corrector scheme (Song et al., 2021). At sampling time $t$, we run a Langevin algorithm for a few iterations targeting $P_t$ (corrector) before performing the jump to another noise level (predictor). Our Algorithm 2 can be interpreted as a "corrector only" scheme, where instead of using the Langevin algorithm, we use MMD flows.

## 7 EXPERIMENTS

**Understanding** DMMD **behavior in 2-D.** Our aim is to get an understanding of the behavior of DMMD described in Section 4. We expect DMMD to mimic GAN discriminator training via noise conditional discriminator learning. To see whether this manifests in practice, we design an experiment with a Radial Basis Function (RBF) kernel for MMD, $k_t(x, y) = \exp[-\|x - y\|^2/(2\sigma^2(t; \theta))]$, where the noise dependent kernel width function $\sigma(\cdot; \theta) : [0, 1] \to [0, +\infty)$ is parameterized by $\theta \in \mathbb{R}^K$. This parameter controls the coarseness of the MMD discriminator. We consider 2-D checkerboard dataset, see Figure 1, left. We learn noise-conditional kernel widths $\sigma(t; \theta)$ using a neural network. As baselines, we train MMD-GAN where the discriminator learns $\sigma$, as well as MMD gradient flow with fixed values of $\sigma$ and with a manually selected noise-dependent $\sigma(t) = 0.1(1 - t) + 0.5t$, i.e. *linear interpolation*. All experimental details are provided in Appendix H.

We report the learned RBF kernel widths for DMMD in Figure 2, left. As expected, as noise level goes from high to low, the kernel width $\sigma(t)$ decreases. In Figure 2, center, we show the learned MMD-GAN kernel width parameter $\sigma$ as a function of training iterations. As the training progresses, this parameter decreases, since the corresponding generator produces samples close to the target distribution. The behaviors of DMMD and MMD-GAN are quite similar and so as the range of values for the kernel widths is also similar. This highlights our point that DMMD mimics the training

of a GAN discriminator. The exact dynamics for $\sigma(t)$ in DMMD depends on the parameters of the forward diffusion process (6). The sharp phase transition is consistent with the phase transition highlighted in Section 3. In addition, we report $\mathrm{MMD}^2(P_t, P; t)$ for different methods in Figure 2, right. We see that DMMD behaves similarly to *linear interpolation*, but is more nuanced for higher noise levels. The samples are reported in Figure 1. DMMD produces samples which are visually better than the other baselines. For the RBF kernel, we noticed the presence of outliers. The amount of outliers generally depends on the kernel, see Appendix of (Hertrich et al., 2024) for more details.

**Image generation** We study the performance of DMMD on unconditional image generation of CIFAR10 (Krizhevsky & Hinton, 2009). We use the same forward diffusion process as in (Ho et al., 2020) to produce noisy images. We use a U-Net (Ronneberger et al., 2015) backbone for discriminator feature network $\phi(x, t; \theta)$, with a slightly different architecture from (Ho et al., 2020), see Appendix I. For all image-based experiments, we use a linear base kernel (11). We explored using other kernels such as RBF and Rational Quadratic (RQ), but did not find an improvement in performance. We use FID (Heusel et al., 2017) and Inception Score (Salimans et al., 2016) for evaluation, see Appendix I. Unless specified otherwise, we use the number $N_{\mathrm{p}} = 200$ of particles for Algorithm 2. We perform ablation over the number of particles in Appenidx I.3. The total number of iterations for DMMD equals to $T \times N_{\mathrm{s}}$, where $T$ is the number of noise levels and $N_{\mathrm{s}}$ is the number of steps per noise level. For consistency with diffusion models, we call this *number of function evaluations* (NFE) and we show performance of DMMD with different NFEs. As we show in Appendix J (see Table 8), there is an improvement on FID as we increase NFEs, but only up to a point (NFE=250).

Table 1: **Unconditional generation on CIFAR-10**. For MMD GAN (orig.), we used mixed-RQ kerned (see (Binkowski et al., 2018)). "Orig." – original paper, "impl." – our implementation. For JKO-Flow (Fan et al., 2022), the NFE is taken from their Figure 12.

| Method | FID | IS | NFE |
|---|---|---|---|
| MMD GAN (orig.) | 39.90 | 6.51 | - |
| MMD GAN (impl.) | 13.62 | 8.93 | - |
| DDPM (orig.) | 3.17 | 9.46 | 1000 |
| DDPM (impl.) | 5.19 | 8.90 | 100 |
| **Discriminator flow baselines** | | | |
| DGGF-KL | 28.80 | - | 110 |
| JKO-Flow | 23.10 | 7.48 | $\sim 150$ |
| **MMD flow baselines** | | | |
| MMD-GAN-Flow | 450 | 1.21 | 100 |
| GS-MMD-RK | 55.00 | - | 86 |
| DMMD (ours) | **8.31** | **9.09** | 100 |
| DMMD (ours) | **7.74** | **9.12** | 250 |

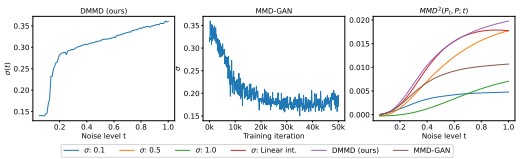

Figure 2: **Toy experiment**. *Left*, learned RBF kernel widths $\sigma(t)$ for DMMD. *Center*, $\sigma$ for MMD-GAN as function of training iterations. *Right*, $\mathrm{MMD}^2(P_t, P; t)$ for different methods.

Table 2: **Approximate sampling** performance on CIFAR10. IS stands for Inception score

| Method | FID | IS | NFE |
|---|---|---|---|
| DMMD | 8.31 | 9.09 | 100 |
| DMMD-$e$ | **8.21** | 8.99 | 102 |
| $a$-DMMD | 24.86 | 9.10 | 50 |
| $a$-DMMD-$e$ | **9.185** | 8.70 | 52 |
| $a$-DMMD-$a$ | 11.22 | 9.00 | 52 |

As baselines, we reimplement MMD-GAN (Binkowski et al., 2018) with linear base kernel and DDPM (Ho et al., 2020) using the same NN backbones as for DMMD. We also report results from the original papers. We further consider baselines based on *discriminator flows*: JKO-Flow (Fan et al., 2022), which uses JKO (Jordan et al., 1998) scheme for the KL gradient flow; and Deep Generative Wasserstein Gradient Flows (DGGF-KL) (Heng et al., 2023), which uses particle-based approach (as in (3)) for the KL gradient flow. These methods use adversarial training to train discriminators (see Section 6). We further compare against Generative Sliced MMD Flows with Riesz Kernels (GS-MMD-RK) (Hertrich et al., 2024) which uses a similar particle-based approach to DGGF-KL to construct MMD flow, but with a fixed (kernel) discriminator. We also report results using a discriminator flow defined on a trained MMD-GAN discriminator, which we call MMD-GAN-Flow. Experimental details are given in Appendix I. The results are provided in Table 1.

We see that DMMD performs better than MMD GAN. As expected, MMD-GAN-Flow does not work. This is because the MMD-GAN discriminator at convergence was trained on samples close to the target distribution. For the RBF kernel experiment, this means that the gradient of MMD will be very small on samples far away from the target distribution. This highlights the benefit of

adaptive MMD discriminators. Moreover, DMMD performs better than GS-MMD-RK, which uses fixed kernel. This highlights the advantage of learning discriminator features in DMMD. DMMD achieves superior performance compared to other discriminator flow baselines. We believe that one of the reasons for the under-performance of these methods is adversarial training, which makes the hyperparameter choice tricky. DMMD, on the other hand, relies on a simple non-adversarial training procedure from Algortihm 1. Finally, we see that DDPM performs better than DMMD. This is not surprising, since both, U-Net architecture and forward diffusion process (6) were optimized for DDPM performance. Nevertheless, DMMD demonstrates strong empirical performance as a discriminator flow method trained without adversarial training. Samples from our method are provided in Appendix L.1. We provide results on CELEB-A, LSUN Church and MNIST below.

**Approximate sampling.** We run approximate MMD gradient flow (see Section 5) with the same discriminator as for DMMD. We call this variant $a$-DMMD, where $a$ stands for *approximate*. We then refine the solution of $a$-DMMD in a "denoising" step by taking two additional MMD gradient flow steps with a higher learning rate, using either the approximate gradient flow, which we call $a$-DMMD-$a$, or the exact gradient flow (9) applied to a single particle, which we call $a$-DMMD-$e$, where $e$ stands for *exact*. Finally, for reference, we add a final denoising step to the original DMMD flow, which we call DMMD-$e$. Results are provided in Table 2. We observe that $a$-DMMD performs worse than DMMD, as expected. Applying a denoising step improves performance of $a$-DMMD, bringing it closer to DMMD. This suggests that the approximation (13) moves the particles close to the target distribution; but once close to the target, a more refined procedure is required. By contrast, we see that denoising helps DMMD only marginally. This suggests that the *exact* noise-conditional witness function (10) accurately captures fine detail close to the target distribution.

**Results on MNIST, CELEB-A (64x64) and LSUN-Church (64x64)** Besides CIFAR-10, we study the performance of DMMD on MNIST (Lecun et al., 1998), CELEB-A (64x64 (Liu et al., 2015) and LSUN-Church (64x64) (Yu et al., 2015). For MNIST and CELEB-A, we consider the same splits and evaluation regime as in (Franceschi et al., 2023). For LSUN Church, the splits and the evaluation regime are taken from (Ho et al., 2020). For more details, see Appendix I.1 — where we also provide results for the high resolution dataset CELEB-A-HQ (128x128), and results on CELEB-A-HQ (256x256) using DMMD in latent space. As baselines, we consider our implementations of DDPM (Ho et al., 2020), MMD-GAN (Binkowski et al., 2018). In addition to DMMD, we report the performance of *Discriminator flow* baseline from (Franceschi et al., 2023) with numbers taken from the corresponding paper. This baseline uses adversarial training together with MMD gradient flow to produce samples. The results are provided in Table 3. We see that DMMD performance is better compared to the discriminator flow and MMD-GAN, which is consistent with our findings on CIFAR-10. It also underperforms compared to DDPM. The corresponding samples are provided in Appendix L.2.

Table 3: **Unconditional generation on additional datasets**. The metric used is FID.

| Dataset | MMD-GAN | DDPM | DMMD | Disc. flow |
|---------|---------|------|------|------------|
| MNIST   | 7.0     | 1.94 | 3.0  | 4.0        |
| CELEB-A | 12.1    | 6.72 | 8.3  | 41.0       |
| LSUN    | 8.4     | 3.84 | 6.1  | -          |

## 8 CONCLUSION

In this paper we have presented a method to train a noise conditional discriminator without adversarial training, using a forward diffusion process. We use this noise conditional discriminator to generate samples using a noise adaptive MMD gradient flow. We provide theoretical insight into why an adaptive gradient flow can provide faster convergence than the non-adaptive variant. We demonstrate strong empirical performance of our method on uncoditional image generation of CIFAR10, as well as on additional, similar image datasets. We propose a scalable approximation of our approach which has close to the original empirical performance.

A number of questions remain open for future work. The empirical performance of DMMD will be of interest in regimes where diffusion models could be ill-behaved, such as in generative modeling on Riemannian manifolds; as well as on larger datasets such as ImageNet. DMMD provides a way of training a discriminator, which may be applicable in other areas where a domain-adaptive discriminator might be required. Finally, it will be of interest to establish theoretical foundations for DMMD in general settings, and to derive convergence results for the associated flow.

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

## A    Organization of the supplementary material

In Appendix B, we briefly describe the adversarial training in the context of Generative Adversarial Networks (GANs). In Appendix C, we describe in details the training and sampling procedures for DMMD. In Appendix D, we provide more details about the computational complexity of DMMD during trianing and sampling. In Appendix E, we explain how our approach could be applied to $f$-divergences resulting in DKALE-Flow and in Appendix F, we provide more details about this method. In Appendix G, we discuss the connection to Flow Matching. In Appendix H, we describe more details for the 2d experiments. In Appendix I, we provide experimental details for the image datasets as well as the additional results. In Appendix K, we provide proof for the theoretical results described in Appendix 3 from the main section of the paper. Finally, in Appendix L we present the samples from DMMD on different image datasets.

## B    Adversarial Training

We briefly describe the notion of the adversarial training in the context of Generative Adversarial Networks (GANs). The objective function for GANs is

$$F(\theta, \psi) = \mathbb{E}_{Z \sim U[-1,1]} D[X; G(Z; \psi); \theta],$$

where $G(Z, \psi)$ is a generator with parameters $\psi$ and $D[X; G(Z; \psi); \theta]$ is a discriminator divergence with parameters $\theta$. The objective for the generator is

$$\psi^* \leftarrow \arg \min_{\psi} \max_{\theta} F(\theta, \psi)$$

The objective for the discriminator is

$$\theta^* \leftarrow \arg \max_{\theta} \min_{\psi} F(\theta, \psi)$$

In practice, we alternate the updates on the objective $F$ for the generator and the discriminator. Let $N$ be the iteration number. Then, the generator parameters are updated as

$$\psi^{n+1} \leftarrow \psi^n - \alpha_{\psi} \nabla_{\psi} F(\theta^n, \psi = \psi^n)$$

and the discriminator parameters are updated as

$$\theta^{n+1} \leftarrow \theta^n + \alpha_{\theta} \nabla_{\theta} F(\theta = \theta^n, \psi^{n+1}),$$

where $\alpha_{\psi}, \alpha_{\theta}$ are learning rates.

When we train DMMD, we could interpret it as having a fixed generator given by the forward diffusion process, and the training is no longer adversarial since there are no min/max optimization going on.

## C    DMMD training and sampling

### C.1    MMD discriminator

Let $\mathcal{X} \subset \mathbb{R}^D$ and $\mathcal{P}(\mathcal{X})$ be the set of probability distributions defined on $\mathcal{X}$. Let $P \in \mathcal{P}(\mathcal{X})$ be the *target* or data distribution and $Q_{\psi} \in \mathcal{P}(\mathcal{X})$ be a distribution associated with a *generator* parameterized by $\psi \in \mathbb{R}^L$. Let $\mathcal{H}$ be Reproducing Kernel Hilbert Space (RKHS), see (Schölkopf & Smola, 2018) for details, for some kernel $k : \mathcal{X} \times \mathcal{X} \to \mathbb{R}$. Maximum Mean Discrepancy (MMD) (Gretton et al., 2012) between $Q_{\psi}$ and $P$ is defined as $\mathrm{MMD}(Q_{\psi}, P) = \sup_{f \in \mathcal{H}} \{ \mathbb{E}_{Q_{\psi}}[f(X)] - \mathbb{E}_P[f(X)] \}$. Given $X^N = \{x_i\}_{i=1}^N \sim Q_{\psi}^{\otimes N}$ and $Y^M = \{y_i\}_{i=1}^M \sim P^{\otimes M}$, an unbiased estimate of $\mathrm{MMD}^2$ (Gretton et al., 2012) is given by

$$\mathrm{MMD}_u^2[X^N, Y^M] = \tfrac{1}{N(N-1)} \sum_{i \neq j}^N k(x_i, x_j) + \tag{14}$$

$$\tfrac{1}{M(M-1)} \sum_{i \neq j}^M k(y_i, y_j) - \tfrac{2}{NM} \sum_{i=1}^N \sum_{j=1}^M k(x_i, y_j).$$

In MMD GAN (Binkowski et al., 2018; Li et al., 2017), the kernel in the objective (14) is given as

$$k(x, y) = k_{\text{base}}(\phi(x; \theta), \phi(y; \theta)), \tag{15}$$

where $k_{\text{base}}$ is a base kernel and $\phi(\cdot; \theta) : \mathcal{X} \to \mathbb{R}^K$ are neural networks *discriminator* features with parameters $\theta \in \mathbb{R}^H$. We use the modified notation of $\text{MMD}_u^2[X^N, Y^M; \theta]$ for equation (14) to highlight the functional dependence on the discriminator parameters. MMD is an instance of Integral Probability Metric (IPM) (see (Arjovsky et al., 2017)) which is well defined on distributions with disjoint support unlike f-divergences (Nowozin et al., 2016). An advantage of using MMD over other IPMs (see for example, Wasserstein GAN (Arjovsky et al., 2017)) is the flexibility to choose a kernel $k$. Another form of MMD is expressed as a norm of a *witness function*

$$\text{MMD}(Q_\psi, P) = \sup_{f \in \mathcal{H}} \{\mathbb{E}_{Q_\psi}[f(X)] - \mathbb{E}_P[f(X)]\} = \|f_{Q_\psi, P}\|_{\mathcal{H}},$$

where the witness function $f_{Q_\psi, P}$ is given as

$$f_{Q_\psi, P}(z) = \int k(x, z) dQ_\psi - \int k(y, z) dP(y)$$

Given two sets of samples $X^N = \{x_i\}_{i=1}^N \sim Q_\psi^{\otimes N}$ and $Y^M = \{y_i\}_{i=1}^M \sim P^{\otimes M}$, and the kernel (15), the empirical witness function is given as

$$\hat{f}_{Q_\psi, P}(z) = \frac{1}{N} \sum_{i=1}^N k_{\text{base}}(\phi(x_i; \theta), \phi(z; \theta)) - \frac{1}{M} \sum_{j=1}^M k_{\text{base}}(\phi(y_j; \theta), \phi(z; \theta))$$

The $\ell_2$ penalty (Binkowski et al., 2018) is defined as

$$\mathcal{L}_{\ell_2}(\theta) = \frac{1}{N} \sum_{i=1}^N \|\phi(x_i; \theta)\|_2^2 + \frac{1}{N} \sum_{i=1}^N \|\phi(y_i; \theta)\|_2^2$$

Assuming that $M = N$ and following (Binkowski et al., 2018; Gulrajani et al., 2017), for $\alpha_i \sim U[0, 1]$, where $U[0, 1]$ is a uniform distribution on $[0, 1]$, we construct $z_i = x_i \alpha_i + (1 - \alpha) y_i$ for all $i = 1, \ldots, N$. Then, the gradient penalty (Binkowski et al., 2018; Gulrajani et al., 2017) is defined as

$$\mathcal{L}_\nabla(\theta) = \frac{1}{N} \sum_{i=1}^N (\|\nabla \hat{f}_{Q_\psi, P}(z_i)\|_2 - 1)^2$$

We denote by $\mathcal{L}(\theta)$ the MMD discriminator loss given as

$$\mathcal{L}(\theta) = -\text{MMD}_u^2[X^N, Y^M; \theta] = \frac{1}{N(N-1)} \sum_{i \neq j}^N k_{\text{base}}(\phi(x_i; \theta), \phi(x_j; \theta)) +$$

$$\frac{1}{M(M-1)} \sum_{i \neq j}^M k_{\text{base}}(\phi(y_i; \theta), \phi(y_j; \theta)) - \frac{2}{NM} \sum_{i=1}^N \sum_{j=1}^M k_{\text{base}}(\phi(x_i; \theta), \phi(y_j; \theta))$$

Then, the total loss for the discriminator on the two samples of data assuming that $N = M$ is given as

$$\mathcal{L}_{\text{tot}}(\theta) = \mathcal{L}(\theta) + \lambda_\nabla \mathcal{L}_\nabla(\theta) + \lambda_{\ell_2} \mathcal{L}_{\ell_2}(\theta),$$

for some constants $\lambda_\nabla \geq 0$ and $\lambda_{\ell_2} \geq 0$.

## C.2 NOISE-DEPENDENT MMD

In Section 4, we describe the approach to train MMD discriminator from forward diffusion using noise-dependent discriminators. For that, we assume that we are given a noise level $t \sim U[0, 1]$ where $U[0, 1]$ is a uniform distribution on $[0, 1]$, and a set of clean data $X^N = \{x^i\}_{i=1}^N \sim P^{\otimes N}$. Then we produce a set of noisy samples $x_t^i$ using forward diffusion process (6). We denote these samples by $X_t^N = \{x_t^i\}_{i=1}^N$. We define noise conditional kernel

$$k(x, y; t, \theta) = k_{\text{base}}(\phi(x, t; \theta), \phi(y, t; \theta)),$$

with noise conditional features $\phi(x, t; \theta)$. This allows us to define the noise conditional discriminator loss

$$\mathcal{L}(\theta, t) = -\text{MMD}_u^2[X^N, X_t^N, t, \theta] = \frac{1}{N(N-1)} \sum_{i \neq j}^{N} k_{\text{base}}(\phi(x_t^i; t, \theta), \phi(x_t^j; t, \theta)) + \tag{16}$$

$$\frac{1}{N(N-1)} \sum_{i \neq j}^{N} k_{\text{base}}(\phi(x^i; t, \theta), \phi(x^j; t, \theta))$$

$$-\frac{2}{N^2} \sum_{i=1}^{N} \sum_{j=1}^{N} k_{\text{base}}(\phi(x^i; t, \theta), \phi(x_t^j; t, \theta))$$

The noise conditional $\ell_2$ penalty is given as

$$\mathcal{L}_{\ell_2}(\theta, t) = \frac{1}{N} \sum_{i=1}^{N} \|\phi(x_t^i; t, \theta)\|_2^2 + \frac{1}{N} \sum_{i=1}^{N} \|\phi(x^i; t, \theta)\|_2^2$$

The noise conditional gradient penalty is given as

$$\mathcal{L}_{\nabla}(\theta, t) = \frac{1}{N} \sum_{i=1}^{N} (\|\nabla \hat{f}_{P,t}(z_i)\|_2 - 1)^2,$$

where $z_i = \alpha_i x_t^i + (1 - \alpha_i) x^i$ for $\alpha_i \sim U[0, 1]$ and the noise conditional witness function

$$\hat{f}_{P,t}(z) = \frac{1}{N} \sum_{i=1}^{N} k_{\text{base}}(\phi(x_i^t; t, \theta), \phi(z; \theta)) - \frac{1}{N} \sum_{j=1}^{N} k_{\text{base}}(\phi(x_i; t, \theta), \phi(z; \theta)) \tag{17}$$

Therefore, the total noise conditional loss is given as

$$\mathcal{L}_{\text{tot}}(\theta, t) = \mathcal{L}(\theta, t) + \lambda_{\nabla} \mathcal{L}_{\nabla}(\theta, t) + \lambda_{\ell_2} \mathcal{L}_{\ell_2}(\theta, t), \tag{18}$$

for some constants $\lambda_{\nabla} \geq 0$ and $\lambda_{\ell_2} \geq 0$.

## C.3 Linear kernel for scalable MMD

Computational complexity of (18) is $O(N^2)$. Here, we assume that the base kernel is linear, i.e.

$$k_{\text{base}}(x, y) = \langle x, y \rangle$$

This allows us to simplify the MMD computation (16) as

$$\text{MMD}_u^2[X^N, X_t^N, t, \theta] = \frac{1}{N(N-1)} \left( \bar{\phi}_t(X_t^N)^T \bar{\phi}_t(X_t^N) - \|\bar{\phi}_t\|^{2^N}(X_t) \right) +$$

$$\frac{1}{N(N-1)} \left( \bar{\phi}_t(X^N)^T \bar{\phi}_t(X^N) - \|\bar{\phi}_t\|^{2^N}(Y) \right) - \frac{2}{NN} (\bar{\phi}_t(X_t^N))^T \bar{\phi}_t(X^N), \tag{19}$$

where

$$\bar{\phi}_t(X_t^N) = \sum_{i=1}^{N} \phi(x_t^i; \theta_t)$$

$$\bar{\phi}_t(X^N) = \sum_{j=1}^{N} \phi(x^i; \theta_t)$$

$$\|\bar{\phi_t}\|^2(X_t^N) = \sum_{i=1}^{N} \|\phi(x_t^i; \theta_t)\|^2$$

$$\|\bar{\phi_t}\|^2(X^N) = \sum_{j=1}^{N} \|\phi(x^i; \theta_t)\|^2$$

---

**Algorithm 3** Approximate noise-adaptive MMD gradient flow for a single particle

---
**Inputs:** $T$ is the number of noise levels
$t_{\max}, t_{\min}$ are maximum and minimum noise levels
$N_s$ is the number of gradient flow steps per noise level
$\eta > 0$ is the gradient flow learning rate
$\bar{\phi}(X_0, t, \theta^\star)$ - precomputed clean features for all $t = 1, \ldots, T$ with (20)
$\bar{\phi}(X_t, t, \theta^\star)$ - precomputed noisy features for all $t = 1, \ldots, T$ with (20)
**Steps:** Sample initial noisy particle $Z \sim N(0, \mathrm{Id})$
**for** $i = T$ **to** $0$ **do**
   Set the noise level $t = i\Delta t$ and $Z_0^t = Z$
   **for** $n = 0$ **to** $N_s - 1$ **do**
      $Z_{n+1}^t = Z_n^t - \eta\langle\nabla_z\phi(Z_n^t, t; \theta^\star), \bar{\phi}(X_t, t, \theta^\star) - \bar{\phi}(X_0, t, \theta^\star)\rangle$
   **end for**
   Set $Z = Z_N^t$
**end for**
Output $Z$

---

Therefore we can pre-compute quantities $\bar{\phi}_t(X_t^N), \bar{\phi}_t(X^N), \|\bar{\phi_t}\|^2(X_t^N), \|\bar{\phi_t}\|^2(X^N)$ which takes $O(N)$ and compute $\mathrm{MMD}_u^2[X^N, X_t^N, t, \theta]$ in $O(1)$ time. This also leads $O(1)$ computation complexity for $\mathcal{L}_{\ell_2}$ and $O(N)$ complexity for $\mathcal{L}_\nabla$. This means that we simplify the computational complexity to $O(N)$ from $O(N^2)$.

At sampling, following (9) requires to compute the witness function (17) for each particle, which for a general kernel takes $O(N^2)$ in total. Using the linear kernel above, simplifies the complexity of the witness as follows

$$\hat{f}_{P,t}(z) = \langle\bar{\phi}_t(Z^N) - \bar{\phi}_t(X^N), \phi(z;\theta)\rangle,$$

where $Z^N$ is a set of $N$ noisy particles. We can precompute $\bar{\phi}_t(Z^N)$ in $O(N)$ time. Therefore one iteration of a witness function will take $O(1)$ time and for $N$ noisy particles it makes $O(N)$.

### C.4 APPROXIMATE SAMPLING PROCEDURE

In this section we provide an algorithm for the approximate sampling procedure. The only change with the original Algorithm 2 is the approximate witness function

$$\hat{f}_{P_t,P}^\star(z) = \langle\phi(z, t; \theta^\star), \bar{\phi}(X_t, t, \theta^\star) - \bar{\phi}(X_0, t, \theta^\star)\rangle,$$

where

$$\bar{\phi}(X_0, t, \theta^\star = \frac{1}{N}\sum_{i=1}^N \phi(x_0^i, t; \theta^\star) \tag{20}$$

$$\bar{\phi}(X_t, t, \theta^\star = \frac{1}{N}\sum_{i=1}^N \phi(x_t^i, t; \theta^\star)$$

Here $x_0^i, i = 1, \ldots, N$ correspond to the whole training set of clean samples and $x_t^i, i = 1, \ldots,$ correspond to the noisy version of these clean samples produced by the forward diffusion process (6) for a given noise level $t$. These features can be precomputed once for every noise level $t$. The resulting algorithm is given in Algorithm (3). A second crucial difference with the original algorithm is that it generates single particles $Z$ independently.

## D COMPUTATIONAL COMPLEXITY

### D.1 TRAINING-TIME COMPLEXITY

Computing total loss in training iteration (see Algorithm 1) on a batch of $B$ samples is $O(B^2)$ for an arbitrary kernel and $O(B)$ for a linear kernel (see Appendix C.3). We compute it for $N_n$ noise levels meaning that the cost scales as $O(N_n B^2)$ for an arbitrary kernel and $O(N_n B)$ for a linear kernel. This is more expensive than DDPM (Ho et al., 2020) which cost scales as $O(B)$.

## D.2 SAMMPLING-TIME COMPLEXITY

During sampling time (see Algorithm 2), we use $T$ noise levens and do $N_s$ steps per noise level with $N_p$ noisy particles. Let $O(F)$ denote the cost of a forward pass for the backbone network. The gradient takes $O(2F)$ to compute. Each sampling step for DDPM (Ho et al., 2020) costs $O(F)$, while one sampling step for DMMD costs $O(FN_p^2)$ for an arbitrary kernel, $O(FN_p)$ for a linear kernel and $O(F)$ when using approximate sampling procedure.

## E  F-DIVERGENCES

The approach described in Section 4 can be applied to any divergence which has a well defined Wasserstein Gradient Flow described by a gradient of the associated witness function. Such divergences include the variational lower bounds on f-divergences, as described by (Nowozin et al., 2016), which are popular in GAN training, and were indeed the basis of the original GAN discriminator (Goodfellow et al., 2014). One such f-divergence is the KL Approximate Lower bound Estimator (KALE, Glaser et al., 2021). Unlike the original KL divergence, which requires a density ratio, the KALE remains well defined for distributions with non-overlapping support. Similarly to MMD, the Wasserstein Gradient of KALE is given by the gradient of a learned witness function. Thus, we train noise-conditional KALE discriminator and use corresponding noise-conditional Wasserstein gradient flow, as with DMMD. We call this method *Diffusion* KALE *flow* (D-KALE-Flow). This approach is described in Appendix F. We found this approach to lead to reasonable empirical results, however unlike with DMMD, it achieved worse performance than a corresponding GAN, see Appendix J.1.

## F  D-KALE-FLOW

In this section, we describe the DKALE-flow algorithm mentioned in Section E. Let $\mathcal{X} \subset \mathbb{R}^D$ and $\mathcal{P}(\mathcal{X})$ be the set of probability distributions defined on $\mathcal{X}$. Let $P \in \mathcal{P}(\mathcal{X})$ be the *target* or data distribution and $Q \in \mathcal{P}(\mathcal{X})$ be some distribution. The KALE objective (see (Glaser et al., 2021)) is defined as

$$KALE(Q,P|\lambda) = (1+\lambda)\max_{h\in\mathcal{H}}\{1 + \int h dQ - \int e^h dP - \frac{\lambda}{2}||h||_{\mathcal{H}}^2\}, \qquad (21)$$

where $\lambda \geq 0$ is a positive constant and $\mathcal{H}$ is the RKHS with a kernel $k$. In practice, KALE divergence (21) can be replaced by a corresponding parametric objective

$$KALE(Q,P|\lambda,\theta,\alpha) = (1+\lambda)\left(\int h(X;\theta,\alpha)dQ(X) - \int e^{h(Y;\theta,\alpha)}dP(Y) - \frac{\lambda}{2}||\alpha||_2^2\right), \quad (22)$$

where

$$h(X;\theta,\alpha) = \phi(X;\theta)^T\alpha,$$

with $\phi(X;\theta) \in \mathbb{R}^D$ and $\alpha \in \mathbb{R}^D$. The objective function (22) can then be maximized with respect to $\theta$ and $\alpha$ for given $Q$ and $P$. Similar to DMMD, we consider a noise-conditional witness function

$$h(x;t,\theta,\alpha,\psi) = \phi(x;t,\theta)^T\alpha(t;\psi)$$

From here, the noise-conditional KALE objective is given as

$$\mathcal{L}(\theta,\psi,t|\lambda) = KALE(P_t,P|\lambda,\theta,\alpha),$$

where $P_t$ is the distribution corresponding to a forward diffusion process, see Section 4. Then, the total noise-conditional objective is given as

$$\mathcal{L}_{\text{tot}}(\theta,\psi,t|\lambda) = \mathcal{L}(\theta,\psi,t|\lambda) + \lambda_\nabla\mathcal{L}_\nabla(\theta,\psi,t) + \lambda_{\ell_2}\mathcal{L}_{\ell_2}(\theta,t),$$

where gradient penalty has similar form to WGAN-GP (Gulrajani et al., 2017)

$$\mathcal{L}_\nabla(\theta,\psi,t) = \mathbb{E}_Z(||\nabla_Z h(Z;t,\theta,\alpha,\psi)||_2 - 1)^2,$$

where $Z = \beta X + (1-\beta)Y$, $\beta \sim U[0,1]$, $X \sim P(X)$ and $Y \sim P(Y)$. The l2 penalty is given as

$$\mathcal{L}_{\ell_2}(\theta,t) = \frac{1}{2}\left(\mathbb{E}_{X\sim P(X)}||\phi(X;t,\theta)||^2 + \mathbb{E}_{Y\sim P(Y)}||\phi(Y;t,\theta)||^2\right)$$

---

**Algorithm 4** Noise-adaptive KALE flow for single particle

---

**Inputs:** $T$ is the number of noise levels
$t_{\max}, t_{\min}$ are maximum and minimum noise levels
$N_{\mathrm{s}}$ is the number of gradient flow steps per noise level
$\eta > 0$ is the gradient flow learning rate
Trained witness function $h(\cdot; t, \theta^\star, \psi^\star)$
**Steps:** Sample initial noisy particle $Z \sim \mathrm{N}(0, \mathrm{Id})$
Set $\Delta t = (t_{\max} - t_{\min})/T$
**for** $i = T$ **to** $0$ **do**
    Set the noise level $t = t_{\min} + i\Delta t$ and $Z_0^t = Z$
    **for** $n = 0$ **to** $N_{\mathrm{s}} - 1$ **do**
        $Z_{n+1}^t = Z_n^t - \eta\nabla h(Z_n^t; t, \theta^\star, \psi^\star)$
    **end for**
    Set $Z = Z_N^t$
**end for**
Output $Z$

---

Therefore, the final objective function to train the discriminator is

$$\mathcal{L}_{\mathrm{tot}}(\theta, \psi | \lambda) = \mathbb{E}_{t \sim U[0,1]} \left[ \mathcal{L}_{\mathrm{tot}}(\theta, \psi, t | \lambda) \right]$$

This objective function depends on RKHS regularization $\lambda$, on gradient penalty regularization $\lambda_\nabla$ and on l2-penalty regularization $\lambda_{\ell_2}$. Unlike in DMMD, we do not use an explicit form for the witness function and do not use the RKHS parameterisation. On one hand, this allows us to have a more scalable approach, since we can compute KALE and the witness function for each individual particle. On the other hand, the explicit form of the witness function in DMMD introduces beneficial inductive bias. In DMMD, when we train the discriminator, we only learn the kernel features, i.e. corresponding RKHS. In D-KALE, we need to learn both, the kernel features $\phi(x; t, \theta)$ as well as the RKHS projections $\alpha(t; \psi)$. This makes the learning problem for D-KALE more complex. The corresponding noise adaptive gradient flow for KALE divergence is described in Algorithm 4. An advantage over original DMMD gradient flow is the ability to run this flow individually for each particle.

## G    CONNECTION TO FLOW MATCHING

Diffusion models (Ho et al., 2020) try to reverse a Gaussian noising dynamic while flow matching (Lipman et al., 2023) methods try to match an interpolation dynamics between the data distribution and a Gaussian distribution. Both methods can be shown to be equivalent given the choice of the schedule (Gao et al., 2024).

Our approach is agnostic to the destruction process and does not rely on an underlying Gaussian assumption of the prior distribution [3]. In particular, one could design destruction processes which are more suitable to the problem at hand. While extensions have been proposed for such destruction processes in the case of Gaussian diffusion (see (Bansal et al., 2023; Daras et al., 2023) for instance) they either rely on a soft Gaussian assumption for the noising process (Daras et al., 2023) or do not enjoy the theoretical properties of our framework (Bansal et al., 2023): at optimality we recover the data distribution, up to the discretization error, regardless of destruction process or prior distribution.

## H    TOY 2-D DATASETS EXPERIMENTS

For the 2-D experiments, we train DMMD using Algorithm (1) for $N_{\mathrm{iter}} = 50000$ steps with a batch size of $B = 256$ and a number of noise levels per batch equal to $N_{\mathrm{noise}} = 128$. The Gradient penalty constant $\lambda_\nabla = 0.1$ whereas the $\ell_2$ penalty is not used. To learn noise-conditional MMD for DMMD, we use a 4-layers MLP $g(t; \theta)$ with ReLU activation to encode $\sigma(t; \theta) = \sigma_{\min} + \mathrm{ReLU}(g(t; \theta))$ with $\sigma_{\min} = 0.001$, which ensures $\sigma(t; \theta) > 0$. The MLP layers have the architecture of $[64, 32, 16, 1]$.

---

[3]Note that this might also not be necessary in the case of flow matching and stochastic interpolant, see (Albergo et al., 2023) for instance

Before passing the noise level $t \in [0, 1]$ to the MLP, we use sinusoidal embedding similar to the one used in (Ho et al., 2020), which produces a feature vector of size 1024. The forward diffusion process from (Ho et al., 2020) have modified parameters such that corresponding $\beta_1 = 10^-4, \beta_T = 0.0002$. On top of that, we discretize the corresponding process using only 1000 possible noise levels, with $t_{\min} = 0.05$ and $t_{\max} = 1.0$. At sampling time for the algorithm 2, we use $t_{\min} = 0.05, t_{\max} = 1.0$, $N_s = 10$ and $T = 100$. The learning rate $\eta = 1.0$. As basleines, we consider MMD-GAN with a generator parameterised by a 3-layer MLP with ELU activations. The architecture of the MLP is $[256, 256, 2]$. The initial noise for the generator is produced from a uniform distribution $U[-1, 1]$ with a dimensionality of 128. The gradient penalty coefficient equals to 0.1. As for the discriminator, the only learnable parameter is $\sigma$. We train MMD-GAN for 250000 iterations with a batch size of $B = 256$. Other variants of MMD gradient flow use the same sampling parameters as DMMD.

We used 1 $a100$ GPU with $40GB$ of memory to run these experiments. In total, all the experiments took less than 2 hours.

## I    IMAGE GENERATION EXPERIMENTS

For the image experiments, we use CIFAR10 (Krizhevsky & Hinton, 2009) dataset. We use the same forward diffusion process as in (Ho et al., 2020). As a Neural Network backbone, we use U-Net (Ronneberger et al., 2015) with a slightly modified architecture from (Ho et al., 2020). Our neural network architecture follows the backbone used in (Ho et al., 2020). On top of that we output the intermediate features at four levels – before down sampling, after down-sampling, before upsampling and a final layer. Each of these feature vectors is processed by a group normalization, the activation function and a linear layer producing an output vector of size 32. To produce the output of a discriminator features, these four feature vectors are concatenated to produce a final output feature vector of size 128. The noise level time is processed via sinusoidal time embedding similar to (Ho et al., 2020). We use a dropout of 0.2. DMMD is trained for $N_{\text{iter}} = 250000$ iterations with a batch size $B = 64$ with number $N_{\text{noise}} = 16$ of noise levels per batch. We use a gradient penalty $\lambda_{\nabla} = 1.0$ and $\ell_2$ regularisation strength $\lambda_{\ell_2} = 0.1$. As evaluation metrics, we use FID (Heusel et al., 2017) and Inception Score (Salimans et al., 2016) using the same evaluation regime as in (Ho et al., 2020). To select hyperparameters and track performance during training, we use FID evaluated on a subset of 1024 images from a training set of CIFAR10.

For CIFAR10, we use random flip data augmentation.

In DMMD we have two sets of hyperparameters, one is used for training in Algorithm 1 and one is used for sampling in Algorithm 2. During training, we fix the sampling parameters and always use these to select the best set of training time hyperparameters. We use $\eta = 0.1$ gradient flow learning rate, $T = 10$ number of noise levels, $N_{\text{p}} = 200$ number of noisy particles, $N_s = 5$ number of gradient flow steps per noise level, $t_{\min} = 0.001$ and $t_{\max} = 1 - 0.001$. We use a batch of 400 clean particles during training. For hyperparameters, we do a grid search for $\lambda_{\nabla} \in \{0, 0.001, 0.01, 0.1, 1.0, 10.0\}$, for $\lambda_{\ell_2} \in \{0, 0.001, 0.01, 0.1, 1.0, 10.0\}$, for dropout rate $\{0, 0.1, 0.2, 0.3\}$, for batch size $\{16, 32, 64\}$. To train the model, we use the same optimization procedure as in (Ho et al., 2020), notably Adam (Kingma & Ba, 2015) optimizer with a learning rate 0.0001. We also swept over the the dimensionality of the output layer $32, 64, 128$, such that each of four feature vectors got the equal dimension. Moreover, we swept over the number of channels for U-Net $\{32, 64, 128\}$ (the original one was 32) and we found that 128 gave us the best empirical results.

After having selected the training-time hyperparameters and having trained the model, we run a sweep for the sampling time hyperparameters over $\eta \in \{1, 0.5, 0.1, 0.04, 0.01\}$, $T \in \{1, 5, 10, 50\}$, $N_s \in \{1, 5, 10, 50\}$, $t_{\min} \in \{0.001, 0.01, 0.1, 0.2\}$, $t_{\max} \in \{0.9, 1 - 0.001\}$. We found that the best hyperparameters for DMMD were $\eta = 0.1$, $N_s = 10$, $T = 10$, $t_{\min} = 0.1$ and $t_{\max} = 0.9$. On top of that, we ran a variant for DMMD with $T = 50$ and $N_s = 5$.

For $a$-DMMD method, we used the same pretrained discriminator as for DMMD but we did an additional sweep over sampling time hyperparameters, because in principle these could be different. We found that the best hyperparameters for $a$-DMMD are $\eta = 0.04$, $t_{\min} = 0.2$, $t_{\max} = 0.9$, $T = 5$, $N_s = 10$.

For the denoising step, see Table 2, for DMMD-$e$, we used 2 steps of DMMD gradient flow with a higher learning rate $\eta^\star = 0.5$ with $t_{\max} = 0.1$ and $t_{\min} = 0.001$. For $a$-DMMD-$e$, we used 2 steps of DMMD gradient flow with a higher learning rate of $\eta^\star = 0.5$ with $t_{\max} = 0.2$ and $t_{\min} = 0.001$. For $a$-DMMD-$e$, we used 2 steps of DMMD gradient flow with a higher learning rate of $\eta^\star = 0.1$ with $t_{\max} = 0.2$ and $t_{\min} = 0.001$. The only parameter we swept over in this experiment was this higher learning rate $\eta^\star$.

After having found the best hyperparameters for sampling, we run the evaluation to compute FID on the whole CIFAR10 dataset using the same regime as described in (Ho et al., 2020).

For MMD-GAN experiment, we use the same discriminator as for DMMD but on top of that we train a generator using the same U-net architecture as for DMMD with an exception that we do not use the 4 levels of features. We use a higher gradient penalty of $\lambda_\nabla = 10$ and the same $\ell_2$ penalty $\lambda_{\ell_2} = 0.1$. We use a batch size of $B = 64$ and the same learning rate as for DMMD. We use a dropout of $0.2$. We train MMD-GAN for 250000 iterations. For each generator update, we do 5 discriminator updates, following (Brock et al., 2019).

For MMD-GAN-Flow experiment, we take the pretrained discriminator from MMD-GAN and run a gradient flow of type (4) on it, starting from a random noise sampled from a Gaussian. We swept over different parameters such as learning rate $\eta$ and number of iterations $N_{\text{iter}}$. We found that none of our parameters led to any reasonable performance. The results in Table 1 are reported using $\eta = 0.1$ and $N_{\text{iter}} = 100$.

## I.1 ADDITIONAL DATASETS

### I.1.1 RESULTS ON CELEB-A, LSUN-CHURCH, MNIST

We study performance of DMMD on additional datasets, MNIST (Lecun et al., 1998), on CELEB-A (64x64 (Liu et al., 2015), on LSUN-Church (64x64) (Yu et al., 2015). For MNIST and CELEB-A, we use the same training/test split as well as the evaluation protocol as in (Franceschi et al., 2023). For LSUN-Church, For LSUN Church, we compute FID on 50000 samples similar to DDPM (Ho et al., 2020). For MNIST, we used the same hyperparameters during training and sampling as for CIFAR-10 with NFE=100, see Appendix I. For CELEB-A and LSUN, we ran a sweep over $\lambda_{\ell_2} \in \{0, 0.001, 0.01, 0.1, 1.0, 10.0\}$ and found that $\ell_2 = 0.001$ led to the best results. For sampling, we used the same parameters as for CIFAR-10 with NFE=100. We use the same architecture for DMMD as for CIFAR-10 experiments. The reported results in Table 4 are given with NFE=100.

The results are provided in Table 4. In addition to DMMD, we report the performance of the *Discriminator flow* baseline from (Franceschi et al., 2023) with numbers taken from the corresponding paper. We see that DMMD performance is significantly better compared to the discriminator flow, which is consistent with our findings on CIFAR-10. The corresponding samples are provided in Appendix L.2.

Table 4: **Unconditional image generation on additional datasets**. The metric used is FID. The number of gradient flow steps for DMMD is 100 and NFE for DDPM is 100.

| Dataset | $MMD$-GAN | DDPM | DMMD | Disc. flow (Franceschi et al., 2023) |
|---|---|---|---|---|
| CELEB-A | 12.1 | 6.72 | 8.3 | 41.0 |
| LSUN-Church | 8.4 | 3.84 | 6.1 | - |
| MNIST | 7.0 | 1.94 | 3.0 | 4.0 |

### I.1.2 RESULTS ON CELEB-A-HQ

We study performance of DMMD on a high-resolution dataset CELEB-A-HQ (128x128) which contains 30000 samples. During training, we track performance on a subset of 2048 samples. We use the same backbone architecture as in DDPM paper (Ho et al., 2020) applied to CELEB-A-HQ (256x256) dataset. For DMMD, we ran a sweep over $\lambda_{\ell_2} \in \{0, 0.001, 0.01, 0.1, 1.0, 10.0\}$ and found that $\ell_2 = 0.001$ led to the best results. At sampling time, we use $T = 10$ noise levels with $N_s = 10$ and $\eta = 0.001$ gradient flow learning rate and with $N_p = 50$ noisy particles (see Algorithm 2 for

more details). We use the same architecture for DMMD as for CIFAR-10 experiments. For both methods, NFE=100.

The results are reported in Table 5. In addition to DMMD, we also train DDPM (Ho et al., 2020) and report the corresponding performance. The results suggest that DMMD still has a performance gap compared to DDPM as observed for lower resolution datasets. The corresponding samples are provided in Appendix L.3.

Table 5: **Unconditional image generation on CELEB-A-HQ (128x128)**. The metric used is FID. The number of gradient flow steps for DMMD is 100 and we use NFE=100 for DDPM.

| DDPM | DMMD |
|------|------|
| 11.67 | 14.09 |

### I.1.3 RESULTS ON LATENT-SPACE CELEB-A-HQ

For memory efficient sample generation in higher dimensions, we study performance of DMMD in the latent space of a VQ-VAE (van den Oord et al., 2017) trained on CELEB-A-HQ (256x256). We follow the approach of the latent diffusion paper (Rombach et al., 2022), and train a VQ-VAE of the corresponding LDM-4 architecture which produces a latent space of the shape $(64 \times 64 \times 3)$. Our baseline for comparison is a latent diffusion model with the LDM-4 architecture. We trained DMMD on the same latent space, where we swept over $\lambda_{\ell_2} \in \{0.05, 0.1, 0.2\}$, and found that $\ell_2 = 0.2$ led to the best results. We used a gradient penalty $\ell_\nabla = 1$, and the same architecture for DMMD as for CIFAR-10 experiments. At sampling time, we used $T = 10$ noise levels with $N_s = 10$ steps per noise level, a gradient flow learning rate $\eta = 0.1$, and $N_p = 200$ noisy particles (see Algorithm 2 for details). The results are given in 6, with NFE=100 for both methods. The corresponding samples are provided in Appendix L.4.

Table 6: **Unconditional image generation on latent space (64x64x3) CELEB-A-HQ (256x256)**. The metric used is FID. The number of gradient flow steps for DMMD is 100 and NFE=100 for DDPM.

| Latent Diffusion | Latent DMMD |
|------------------|-------------|
| 5.6 | 8.9 |

### I.2 D-KALE-FLOW DETAILS

We study performance of D-KALE-flow on CIFAR10. We use the same architectural setting as in DMMD with the only difference of adding an additional mapping $\alpha(t; \psi)$ from noise level to $D$ dimensional feature vector, which is represented by a 2 layer MLP with hidden dimensionality of 64 and GELU activation function. We use batch size $B = 256$, dropout rate equal to 0.3. For sampling time parameters during training, we use $\eta = 0.5$, total number of noise levels $T = 20$, and number of steps per noise level $N_s = 5$. At training, we sweep over RKHS regularization $\lambda \in \{0, 1, 10, 100, 500, 1000, 2000\}$, gradient penalty $\lambda_\nabla \in \{0, 0.1, 1.0, 10.0, 50.0, 100.0, 250.0, 500.0, 1000.0\}$, l2 penalty in $\{0, 0.1, 0.01, 0.001\}$.

### I.3 NUMBER OF PARTICLES ABLATION

**Number of particles.** In Table 7 we report performance of DMMD depending on the number of particles $N_p$ at sampling time. As expected as the number of particles increases, the FID score decreases, but the overall performance is sensitive to the number of particles. This motivates the approximate sampling procedure from Section 5.

## J PERFORMANCE VS. NUMBER OF GRADIENT FLOW STEPS TRADE-OFF

Here, we provide a table showing the dependence of the performance of DMMD on number of total DMMD gradient flow steps, which we call NFE. The NFE is the total number of gradient flow

Table 7: **Number of particles ablation**, FIDs on CIFAR10.

| $N_\mathrm{p} = 50$ | $N_\mathrm{p} = 100$ | $N_\mathrm{p} = 200$ |
|---|---|---|
| 9.76 | 8.55 | 8.31 |

iterations, which equals to $N_s T$, where $N_s$ is the number of steps per noise level and $T$ is the number of noise levels. By default, we use the gradient flow learning rate $\eta = 0.1$, see (9). We also found that as we increase the number of total gradient flow steps, it was sometimes beneficial to use a smaller learning rate, $\eta = 0.05$. Results are given in Table 8. We see that as we increase NFE, the FID improves up to a point (NFE = 250). After NFE=250, we do not see a further improvement. Moreover, as we noticed in our experiments, increasing the total compute at sampling time might require readjusting the gradient flow learning rate.

Table 8: Dependence of the FID on CIFAR-10 on the total number of gradient flow steps (NFE). $\eta$ is the gradient flow learning rate, see (9).

| Total number of steps (NFE) | FID |
|---|---|
| $10(\eta = 0.1)$ | 377.5 |
| $50(\eta = 0.1)$ | 36.4 |
| $100(\eta = 0.1)$ | 8.5 |
| $250(\eta = 0.1)$ | 12.1 |
| $250(\eta = 0.05)$ | 7.74 |
| $500(\eta = 0.05)$ | 8.6 |
| $1000(\eta = 0.05)$ | 9.1 |

### J.1 RESULTS WITH F-DIVERGENCE

We study performance of D-KALE-Flow described in Section E and Appendix F, in the setting of unconditional image generation for CIFAR-10. We compare against a GAN baseline which uses the KALE divergence in the discriminator, but has adversarially trained generator. More details are described in Appendix F and Appendix I.2. The results are given in Table 9. We see that unlike with DMMD, D-KALE-Flow achieves worse performance than corresponding KALE-GAN - indicating that the inductive bias provided by the generator may be more helpful in this case - this is a topic for future study.

Table 9: **Unconditional image generation on CIFAR-10** with KALE-divergence. The number of gradient flow steps is 100.

| Method | FID | Inception score |
|---|---|---|
| D-KALE-Flow | 15.8 | 8.5 |
| KALE-GAN | 12.7 | 8.7 |

### J.2 COMPUTE RESOURCES FOR IMAGE EXPERIMENTS

For all the experiments, we used $A100$ GPUs with $40$ GB of memory. To train DMMD for $250k$ steps, we needed to run training for around 24 hours. The total hyperparameter sweep for DMMD required 36 runs to figure out regularization constants, 12 runs to figure out batch size and dropout rate and then 3 runs to figure out the dimensionality of the U-Net and the same 3 runs where the features of the U-Net were coming only from the last layer. This required 54 runs in total.

Running inference on small subset of CIFAR-10 required around 2 minutes of GPU time, and we ran full grid search to select best sampling time parameters, which is around $1080$ values. We did this sweep for DMMD and $a - $ DMMD. For DMMD $- e$, we additionally swept over higher learning rate at the second stage which required 5 more runs. For $a - $ DMMD $- e$ and $a - $ DMMD $- a$, we

swept over learning rates at second stage which required 10 more runs. After having found the best parameters, we run sampling with the best parameters on full CIFAR-10 which takes about 1 hour for $NFE = 100$.

For additional datasets, for $MNIST$ we used the same best parameters as for CIFAR-10, which required one run only since we saw very good performance out of the box. For CELEB-A and LSUN, we ran an additional sweep over regularization strength which required 6 training runs per dataset and 2 additional runs for sampling the whole datasets.

For MMD $- GAN$, the training runs were faster, by around 2-x factor. We did a grid search over the regularization strengths which took 36 training runs and 12 runs to figure out batch size and drop-out rate.

For DKALE-flow, the experiment was as fast as MMD $- GAN$ and we ran a grid search with 67 runs for regularization and 4 runs for dropout. The same was done for DKALE $- GAN$.

## K  OPTIMAL KERNEL WITH GAUSSIAN MMD

In this section, we prove the results of Section 3. We consider the following unnormalized Gaussian kernel

$$k_\alpha(x, y) = \alpha^{-d} \exp[-\|x - y\|^2 / (2\alpha^2)].$$

For any $\mu \in \mathbb{R}^d$ and $\sigma > 0$ we denote $\pi_{\mu,\sigma}$ the Gaussian distribution with mean $\mu$ and covariance matrix $\sigma^2\text{Id}$. We denote $\text{MMD}_\alpha^2$ the $\text{MMD}^2$ associated with $k_\alpha$. More precisely for any $\mu_1, \mu_2 \in \mathbb{R}^d$ and $\sigma_1, \sigma_2 > 0$ we have

$$\text{MMD}_\alpha^2(\pi_{\mu_1,\sigma_1}, \pi_{\mu_2,\sigma_2}) = \mathbb{E}_{\pi_{\mu_1,\sigma_1} \otimes \pi_{\mu_1,\sigma_1}}[k_\alpha(X, X')] - 2\mathbb{E}_{\pi_{\mu_1,\sigma_1} \otimes \pi_{\mu_2,\sigma_2}}[k_\alpha(X, Y)] + \quad (23)$$
$$\mathbb{E}_{\pi_{\mu_2,\sigma_2} \otimes \pi_{\mu_2,\sigma_2}}[k_\alpha(Y, Y')].$$

In this section we prove the following result.

**Proposition K.1.** *For any $\mu_0 \in \mathbb{R}^d$ and $\sigma > 0$, let $\alpha^\star$ be given by*

$$\alpha^\star = \text{argmax}_{\alpha \geq 0} \|\nabla_{\mu_0} \text{MMD}_\alpha^2(\pi_{0,\sigma}, \pi_{\mu_0,\sigma})\|.$$

*Then, we have that*

$$\alpha^\star = \text{ReLU}(\|\mu_0\|^2 / (d + 2) - 2\sigma^2)^{1/2}. \quad (24)$$

Before proving Proposition K.1, let us provide some insights on the result. The quantity $\|\nabla_{\mu_0} \text{MMD}_\alpha^2(\pi_{0,\sigma}, \pi_{\mu_0,\sigma})\|$ represents how much the mean of the Gaussian $\pi_{\mu_0,\sigma}$ is displaced when considering a flow on the mean of the Gaussian w.r.t. $\text{MMD}_\alpha^2$. Intuitively, we aim for $\|\nabla_{\mu_0} \text{MMD}_\alpha^2(\pi_{0,\sigma}, \pi_{\mu_0,\sigma})\|$ to be as large as possible as this represents the *maximum displacement possible*. Hence, this justifies our goal of maximizing $\|\nabla_{\mu_0} \text{MMD}_\alpha^2(\pi_{0,\sigma}, \pi_{\mu_0,\sigma})\|$ with respect to the width parameter $\alpha$.

We show that the optimal width $\alpha^\star$ has a closed form given by (24). It is notable that, assuming that $\sigma > 0$ is fixed, this quantity depends on $\|\mu_0\|$, i.e. how far the modes of the two distributions are. This observation justifies our approach of following an *adaptive* MMD flow at inference time. Finally, we observe that there exists a threshold, i.e. $\|\mu_0\|^2 / (d + 2) = 2\sigma^2$ for which lower values of $\|\mu_0\|$ still yield $\alpha^\star = 0$. This phase transition behavior is also observed in our experiments.

We define $\text{D}(\alpha, \sigma, \mu_0, \mu_1)$ for any $\alpha, \sigma > 0$ and $\mu_0, \mu_1 \in \mathbb{R}^d$ given by

$$\text{D}(\alpha, \sigma, \mu_0, \mu_1) = \int_{\mathbb{R}^d \times \mathbb{R}^d} k_\alpha(x, y) \text{d}\pi_{\mu_0,\sigma}(x) \text{d}\pi_{\mu_1,\sigma}(y).$$

**Proposition K.2.** *For any $\alpha, \sigma > 0$ and $\mu_0, \mu_1 \in \mathbb{R}^d$ we have*

$$\text{D}(\alpha, \sigma, \mu_0, \mu_1) = [\alpha^2\sigma^2(1/\kappa^2 + 1/\alpha^2)]^{-d/2} \exp[\|\hat{\mu}_0\|^2 / (2\kappa^2) + \|\hat{\mu}_1\|^2 / (2\kappa^2)$$
$$- \langle \hat{\mu}_0, \hat{\mu}_1 \rangle / \alpha^2 - \|\mu_0\|^2 / (2\sigma^2) - \|\mu_1\|^2 / (2\sigma^2)],$$

*with*

$$\hat{\mu}_1 = (\alpha^2\mu_1 + \kappa^2\mu_0) / (\kappa^2 + \alpha^2),$$
$$\hat{\mu}_0 = (\alpha^2\mu_0 + \kappa^2\mu_1) / (\kappa^2 + \alpha^2),$$

*where $\kappa = (1/\sigma^2 + 1/\alpha^2)^{-1/2}$.*

*Proof.* In what follows, we start by computing $D(\alpha, \sigma, \mu_0, \mu_1)$ for any $\alpha, \sigma > 0$ and $\mu_0, \mu_1 \in \mathbb{R}^d$ given by

$$D(\alpha, \sigma, \mu_0, \mu_1) = \int_{\mathbb{R}^d \times \mathbb{R}^d} k_\alpha(x, y) \mathrm{d}\pi_{\mu_0, \sigma}(x) \mathrm{d}\pi_{\mu_1, \sigma}(y)$$
$$= 1/(2\pi\sigma^2\alpha)^d \int_{\mathbb{R}^d \times \mathbb{R}^d} \exp[-\|x - y\|^2/(2\alpha^2)] \exp[-\|x - \mu_0\|^2/(2\sigma^2)] \exp[-\|y - \mu_1\|^2/(2\sigma^2)] \mathrm{d}x\mathrm{d}y$$
$$= 1/(2\pi\sigma^2\alpha)^d \int_{\mathbb{R}^d \times \mathbb{R}^d} \exp[-\|x - y\|^2/(2\alpha^2) - \|x - \mu_0\|^2/(2\sigma^2) - \|y - \mu_1\|^2/(2\sigma^2)] \mathrm{d}x\mathrm{d}y.$$

In what follows, we denote $\kappa = (1/\sigma^2 + 1/\alpha^2)^{-1/2}$. We have

$$D(\alpha, \sigma, \mu_0, \mu_1) = C(\mu_0, \mu_1) \int_{\mathbb{R}^d \times \mathbb{R}^d} \exp[-\|x\|^2/(2\kappa^2) - \|y\|^2/(2\kappa^2) + \langle x, y\rangle/\alpha^2 + \langle x, \mu_0\rangle/\sigma^2 + \langle y, \mu_1\rangle/\sigma^2] \mathrm{d}x\mathrm{d}y,$$

with $C(\mu_0, \mu_1) = 1/(2\pi\sigma^2\alpha)^d \exp[-\|\mu_0\|^2/(2\sigma^2) - \|\mu_1\|^2/(2\sigma^2)]$. In what follows, we denote $P(x, y)$ the second-order polynomial given by

$$P(x, y) = \|x\|^2/(2\kappa^2) + \|y\|^2/(2\kappa^2) - \langle x, y\rangle/\alpha^2 - \langle x, \mu_0\rangle/\sigma^2 - \langle y, \mu_1\rangle/\sigma^2.$$

Note that we have

$$D(\alpha, \sigma, \mu_0, \mu_1) = C(\mu_0, \mu_1) \int_{\mathbb{R}^d \times \mathbb{R}^d} \exp[-P(x, y)] \mathrm{d}x\mathrm{d}y. \tag{25}$$

Next, for any $\hat\mu_0, \hat\mu_1 \in \mathbb{R}^d$, we consider $Q(x, y)$ given by

$$Q(x, y) = \|x - \hat\mu_0\|^2/(2\kappa^2) + \|y - \hat\mu_1\|^2/(2\kappa^2) - \langle x - \hat\mu_0, y - \hat\mu_1\rangle/\alpha^2$$
$$= \|x\|^2/(2\kappa^2) + \|\hat\mu_0\|^2/(2\kappa^2) + \|y\|^2/(2\kappa^2) + \|\hat\mu_1\|^2/(2\kappa^2) - \langle x, \hat\mu_0\rangle/\kappa^2 - \langle y, \hat\mu_1\rangle/\kappa^2 - \langle x - \hat\mu_0, y - \hat\mu_1\rangle/\alpha^2$$
$$= \|x\|^2/(2\kappa^2) + \|\hat\mu_0\|^2/(2\kappa^2) + \|y\|^2/(2\kappa^2) + \|\hat\mu_1\|^2/(2\kappa^2) - \langle x, \hat\mu_0\rangle/\kappa^2 - \langle y, \hat\mu_1\rangle/\kappa^2$$
$$\quad - \langle x, y\rangle/\alpha^2 - \langle \hat\mu_0, \hat\mu_1\rangle/\alpha^2 + \langle y, \hat\mu_0\rangle/\alpha^2 + \langle x, \hat\mu_1\rangle/\alpha^2$$
$$= P(x, y) + \|\hat\mu_0\|^2/(2\kappa^2) + \|\hat\mu_1\|^2/(2\kappa^2) - \langle x, \hat\mu_0\rangle/\kappa^2 - \langle y, \hat\mu_1\rangle/\kappa^2 + \langle x, \mu_0\rangle/\sigma^2 + \langle y, \mu_1\rangle/\sigma^2$$
$$\quad - \langle \hat\mu_0, \hat\mu_1\rangle/\alpha^2 + \langle y, \hat\mu_0\rangle/\alpha^2 + \langle x, \hat\mu_1\rangle/\alpha^2$$
$$= P(x, y) + \|\hat\mu_0\|^2/(2\kappa^2) + \|\hat\mu_1\|^2/(2\kappa^2) - \langle \hat\mu_0, \hat\mu_1\rangle/\alpha^2$$
$$\quad + \langle x, \mu_0/\sigma^2 - \hat\mu_0/\kappa^2 + \hat\mu_1/\alpha^2\rangle + \langle y, \mu_1/\sigma^2 - \hat\mu_1/\kappa^2 + \hat\mu_0/\alpha^2\rangle.$$

In what follows, we set $\hat\mu_0, \hat\mu_1$ such that

$$\mu_1/\sigma^2 = \hat\mu_1/\kappa^2 - \hat\mu_0/\alpha^2,$$
$$\mu_0/\sigma^2 = \hat\mu_0/\kappa^2 - \hat\mu_1/\alpha^2.$$

We get that

$$\hat\mu_1 = (\mu_1/(\sigma^2\kappa^2) + \mu_0/(\sigma^2\alpha^2))/(1/\kappa^4 - 1/\alpha^4),$$
$$\hat\mu_0 = (\mu_0/(\sigma^2\kappa^2) + \mu_1/(\sigma^2\alpha^2))/(1/\kappa^4 - 1/\alpha^4).$$

We have that

$$\sigma^2(1/\kappa^4 - 1/\alpha^4) = \sigma^2(1/\sigma^4 + 2/(\sigma^2\alpha^2)) = 1/\sigma^2 + 2/\alpha^2 = 1/\kappa^2 + 1/\alpha^2. \tag{26}$$

Therefore, we get that

$$\hat\mu_1 = (\mu_1/\kappa^2 + \mu_0/\alpha^2)/(1/\kappa^2 + 1/\alpha^2),$$
$$\hat\mu_0 = (\mu_0/\kappa^2 + \mu_1/\alpha^2)/(1/\kappa^2 + 1/\alpha^2).$$

Finally, we get that

$$\hat\mu_1 = (\alpha^2\mu_1 + \kappa^2\mu_0)/(\kappa^2 + \alpha^2),$$
$$\hat\mu_0 = (\alpha^2\mu_0 + \kappa^2\mu_1)/(\kappa^2 + \alpha^2).$$

With this choice, we get that

$$P(x, y) = Q(x, y) - \|\hat\mu_0\|^2/(2\kappa^2) - \|\hat\mu_1\|^2/(2\kappa^2) + \langle \hat\mu_0, \hat\mu_1\rangle/\alpha^2 \tag{27}$$

We also have that for any $x, y \in \mathbb{R}^d$

$$Q(x, y) = (1/2) \begin{pmatrix} x - \hat\mu_0 \\ y - \hat\mu_1 \end{pmatrix}^\top \begin{pmatrix} \mathrm{Id}/\kappa^2 & -\mathrm{Id}/\alpha^2 \\ -\mathrm{Id}/\alpha^2 & \mathrm{Id}/\kappa^2 \end{pmatrix} \begin{pmatrix} x - \hat\mu_0 \\ y - \hat\mu_1 \end{pmatrix}$$

Using this result we have that

$$\int_{\mathbb{R}^d \times \mathbb{R}^d} \exp[-Q(x,y)] = (2\pi)^d \det(\Sigma^{-1})^{-1/2}, \tag{28}$$

with

$$\Sigma^{-1} = \begin{pmatrix} \mathrm{Id}/\kappa^2 & -\mathrm{Id}/\alpha^2 \\ -\mathrm{Id}/\alpha^2 & \mathrm{Id}/\kappa^2 \end{pmatrix}.$$

Using (26), we get that

$$\det(\Sigma^{-1}) = [(1/\sigma^2)(1/\kappa^2 + 1/\alpha^2)]^d.$$

Combining this result and (28) we get that

$$\int_{\mathbb{R}^d \times \mathbb{R}^d} \exp[-Q(x,y)] = (2\pi)^d [(1/\sigma^2)(1/\kappa^2 + 1/\alpha^2)]^{-d/2}.$$

Combining this result, (27) and (25) we get that

$$D(\alpha, \sigma, \mu_0, \mu_1) = C(\mu_0, \mu_1) \exp[\|\hat\mu_0\|^2/(2\kappa^2) + \|\hat\mu_1\|^2/(2\kappa^2) - \langle \hat\mu_0, \hat\mu_1 \rangle/\alpha^2](2\pi)^d [(1/\sigma^2)(1/\kappa^2 + 1/\alpha^2)]^{-d/2}.$$

Therefore, we get that

$$D(\alpha, \sigma, \mu_0, \mu_1) = [\alpha^2 \sigma^2 (1/\kappa^2 + 1/\alpha^2)]^{-d/2} \exp[\|\hat\mu_0\|^2/(2\kappa^2) + \|\hat\mu_1\|^2/(2\kappa^2)$$
$$- \langle \hat\mu_0, \hat\mu_1 \rangle/\alpha^2 - \|\mu_0\|^2/(2\sigma^2) - \|\mu_1\|^2/(2\sigma^2)].$$

$\square$

We investigate two special cases of Proposition K.2.

First, we show that if $\mu_0 = \mu_1$ then $D(\alpha, \sigma, \mu_0, \mu_0)$ does not depend on $\mu_0$.

**Proposition K.3.** *For any* $\alpha, \sigma > 0$ *and* $\mu_0 \in \mathbb{R}^d$ *we have* $D(\alpha, \sigma, \mu_0, \mu_0) = (\alpha^2 + 2\sigma^2)^{-d/2}$.

*Proof.* We have that $\hat\mu_0 = \hat\mu_1 = \mu_1 = \mu_0$ in Proposition K.2. In addition, we have that

$$(1/2\kappa^2) + (1/2\kappa^2) - 1/\alpha^2 - 1/(2\sigma^2) - 1/(2\sigma^2) = 0.$$

Therefore, we have that

$$\exp[\|\hat\mu_0\|^2/(2\kappa^2) + \|\hat\mu_1\|^2/(2\kappa^2) - \langle \hat\mu_0, \hat\mu_1 \rangle/\alpha^2 - \|\mu_0\|^2/(2\sigma^2) - \|\mu_1\|^2/(2\sigma^2)] = 1,$$

which concludes the proof upon using that $1/\kappa^2 = 1/\alpha^2 + 1/\sigma^2$. $\square$

Proposition K.3 might seem surprising at first but in fact it simply highlights the fact that when trying to differentiate a Gaussian measure with itself, the result is independent of the location of the Gaussian and only depends on its scale. Then, we study the case where $\mu_1 = 0$.

**Proposition K.4.** *For any* $\alpha, \sigma > 0$ *and* $\mu_0 \in \mathbb{R}^d$ *we have*

$$D(\alpha, \sigma, \mu_0, 0) = (\alpha^2 + 2\sigma^2)^{-d/2} \exp[-\|\mu_0\|^2/(2(\alpha^2 + 2\sigma^2))].$$

*Proof.* First, we have that

$$\hat\mu_0 = \alpha^2/(\kappa^2 + \alpha^2)^2 \mu_0, \qquad \hat\mu_1 = \kappa^2/(\kappa^2 + \alpha^2)^2 \mu_0.$$

Therefore, we get that

$$D(\alpha, \sigma, \mu_0, 0) = [\sigma^2(1/\kappa^2 + 1/\alpha^2)]^{d/2} \exp[(1/2)\{(\alpha^4/\kappa^2 - \kappa^2)/(\kappa^2 + \alpha^2) - 1/\sigma^2\}\|\mu_0\|^2]$$

Using (26) we get that

$$\alpha^4/\kappa^2 - \kappa^2 = \alpha^2(\alpha^2 + \kappa^2)/\sigma^2.$$

Therefore, we get that

$$(\alpha^4/\kappa^2 - \kappa^2)/(\kappa^2 + \alpha^2) - 1/\sigma^2 = (\alpha^2/(\alpha^2 + \kappa^2) - 1)/\sigma^2 = -1/(\alpha^2(1 + 2\sigma^2/\alpha^2)),$$

which concludes the proof. $\square$

Using Proposition K.3, Proposition K.4 and definition (23), we have the following result.

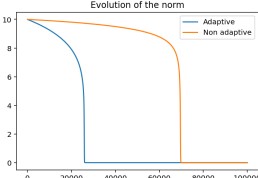 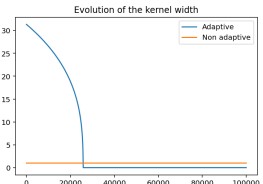

Figure 3: Evolution of the norm of the mean $\mu_t$ of the Gaussian distribution $\pi_{\mu_t,\sigma}$ according to a gradient flow on the mean $\mu_t$ w.r.t. $\mathrm{MMD}_{\alpha_t}$. In the *adaptive* case $\alpha_t$ is given by Proposition 3.1 while in the *non adaptive* case, $\alpha_t = \alpha_0 = 1$. In our experiment we consider $d = 1$ and $\sigma = 1$, for illustration purposes.

**Proposition K.5.** *For any $\alpha, \sigma > 0$ and $\mu_0 \in \mathbb{R}^d$ we have*

$$\mathrm{MMD}^2(\pi_{0,\sigma}, \pi_{\mu_0,\sigma}) = 2(\alpha^2 + 2\sigma^2)^{-d/2}(1 - \exp[-\|\mu_0\|^2/(2(\alpha^2 + 2\sigma^2))]).$$

*In addition, we have*

$$\nabla_{\mu_0}\mathrm{MMD}^2(\pi_{0,\sigma}, \pi_{\mu_0,\sigma}) = -2(\alpha^2 + 2\sigma^2)^{-d/2-1}\exp[-\|\mu_0\|^2/(2(\alpha^2 + 2\sigma^2))]\mu_0.$$

Finally, we have the following proposition.

**Proposition K.6.** *For any $\mu_0 \in \mathbb{R}^d$ and $\sigma > 0$ let $\alpha^\star$ be given by*

$$\alpha^\star = \mathrm{argmax}_{\alpha \geq 0}\|\nabla_{\mu_0}\mathrm{MMD}^2(\pi_{0,\sigma}, \pi_{\mu_0,\sigma})\|.$$

*Then, we have that*

$$\alpha^\star = \mathrm{ReLU}(\|\mu_0\|^2/(d+2) - 2\sigma^2)^{1/2}.$$

*Proof.* Let $\sigma > 0$ and $\mu_0 \in \mathbb{R}^d$. First, using Proposition K.5, we have that for

$$\|\nabla_{\mu_0}\mathrm{MMD}^2(\pi_{0,\sigma}, \pi_{\mu_0,\sigma})\|^2 = 4\alpha^{2d}\|\mu_0\|^2(\alpha^2 + 2\sigma^2)^{-d-2}\exp[-\|\mu_0\|^2/(\alpha^2 + 2\sigma^2)].$$

Next, we study the function $\mathrm{f} : [0, t_0] \to \mathbb{R}$ given for any $t \in [0, t_0]$ by

$$\mathrm{f}(t) = t^{d+2}\exp[-t\|\mu_0\|^2],$$

with $t_0 = 1/(2\sigma^2)$. We have that

$$\mathrm{f}'(t) = t^{d+1}\exp[-t\|\mu_0\|^2]((d+2) - \|\mu_0\|^2 t).$$

We then consider two cases. First, if $t_0 \leq (d+2)/\|\mu_0\|^2$, i.e. $\sigma^2 \leq \|\mu_0\|^2/(2(d+2))$, then f is increasing on $[0, t_0]$ and we have that $f$ is maximum if $t = t_0$. Hence, if $\sigma^2 \leq \|\mu_0\|^2/(2(d+2))$, we have that $\alpha^\star = 0$. Second, if $t_0 \leq (d+2)/\|\mu_0\|^2$, i.e. $\sigma^2 \leq \|\mu_0\|^2/(2(d+2))$ then f is increasing on $[0, t^\star]$, non-increasing on $[t^\star, t_0]$ with $t^\star = (d+2)/\|\mu_0\|^2$ and we have that $f$ is maximum if $t = t^\star$. Hence, if $\sigma^2 \geq \|\mu_0\|^2/(2(d+2))$, we have that $\alpha^\star = (\|\mu_0\|^2/(d+2) - 2\sigma^2)^{1/2}$, which concludes the proof. $\square$

### K.1 PHASE TRANSITION BEHAVIOUR

## L IMAGE GENERATION SAMPLES

### L.1 CIFAR10 SAMPLES

Samples from DMMD with NFE=100 and NFE=250 are given in Figure 4. Samples from DMMD with NFE=100 and from $a$-DMMD with NFE=50 are given in Figure 5.

### L.2 ADDITIONAL DATASETS SAMPLES

Samples for MNIST are given in Figure 6, for CELEB-A (64x64) are given in Figure 7 and for LSUN Church (64x64) are given in Figure 8.

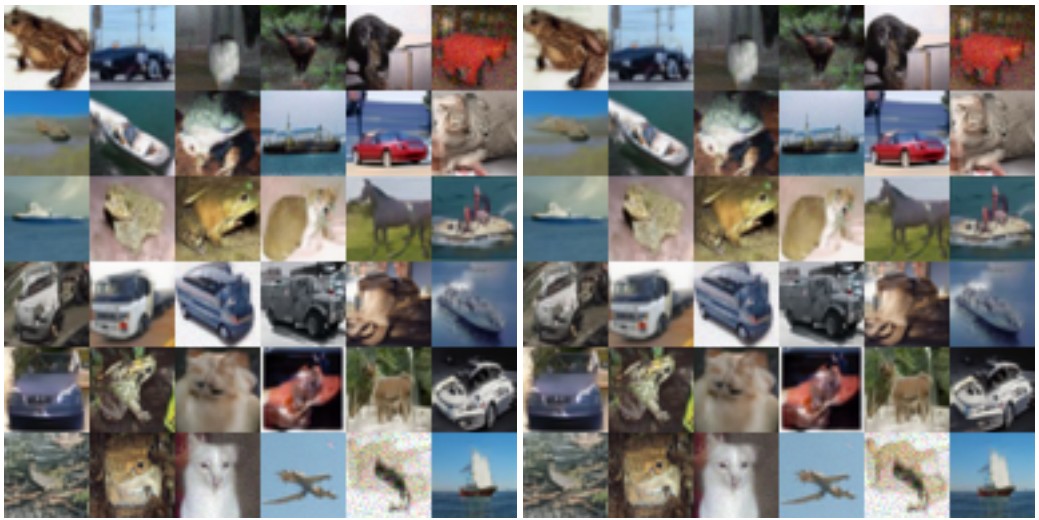

Figure 4: CIFAR-10 samples from DMMD with NFE=250 on the left and with NFE=100 on the right

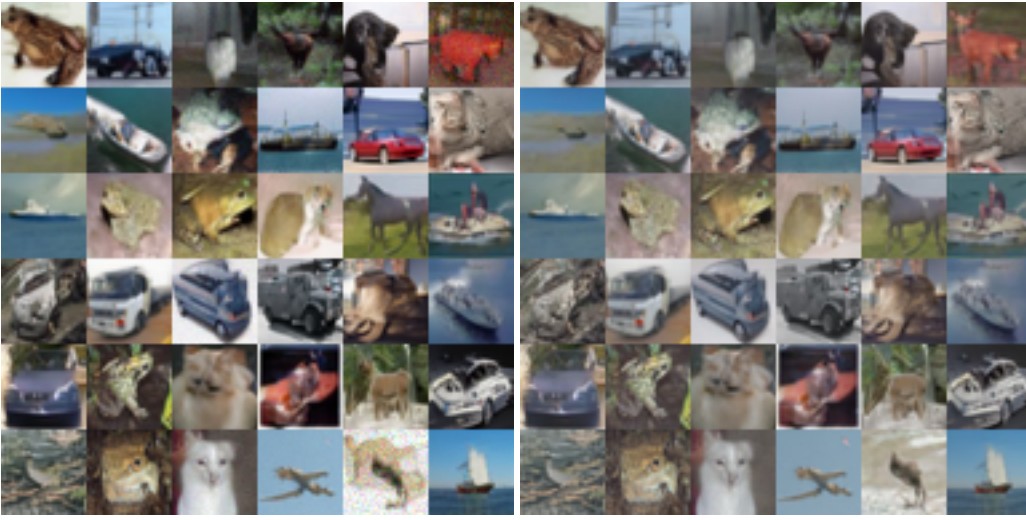

Figure 5: CIFAR-10 samples from DMMD with NFE=100 on the left and samples from the $a$-DMMD-$e$ with NFE=50 on the right

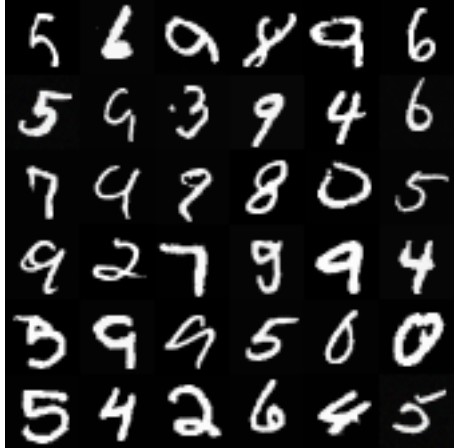

Figure 6: DMMD samples for MNIST.

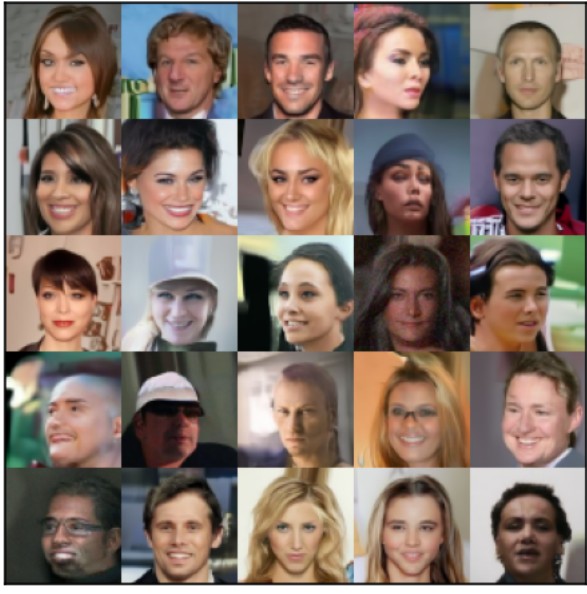

Figure 7: DMMD samples for CELEB-A (64x64).

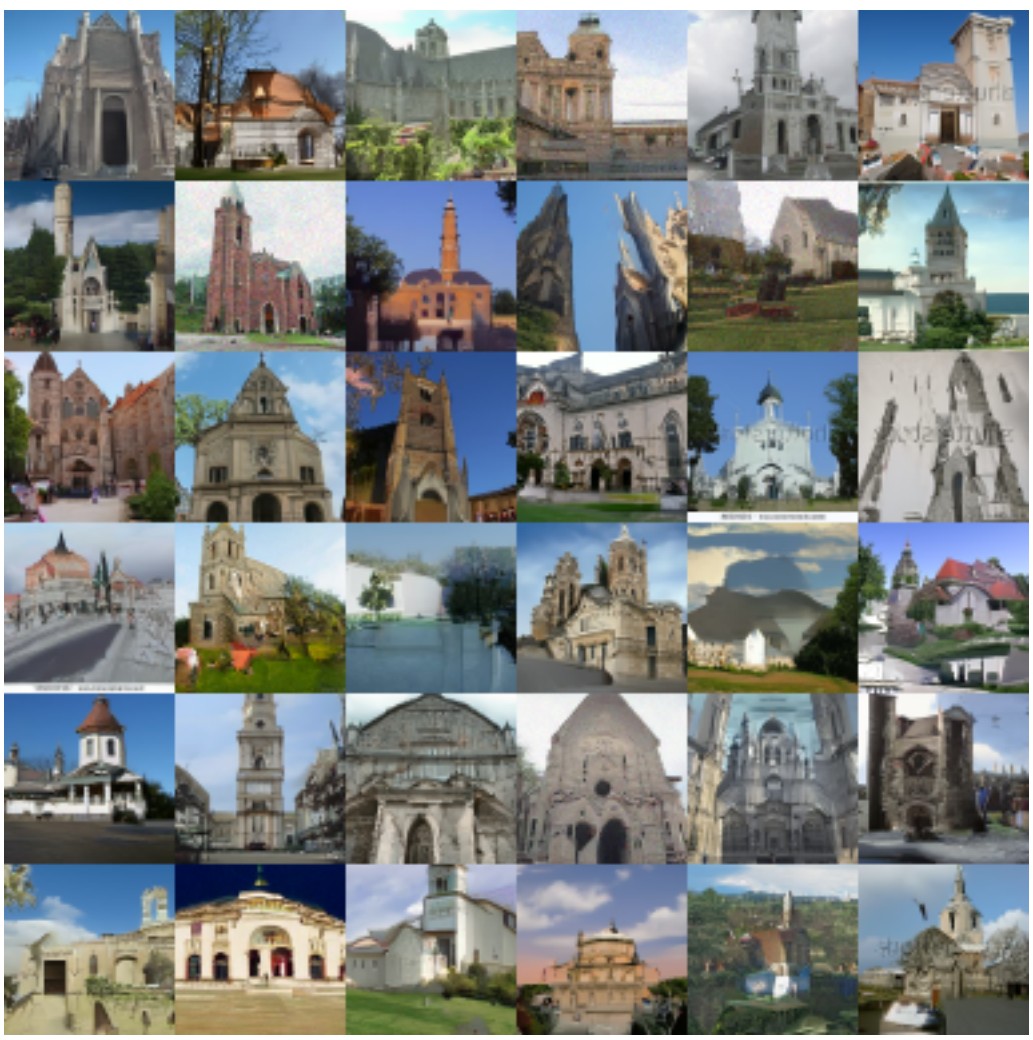

Figure 8: DMMD samples for LSUN Church (64x64).

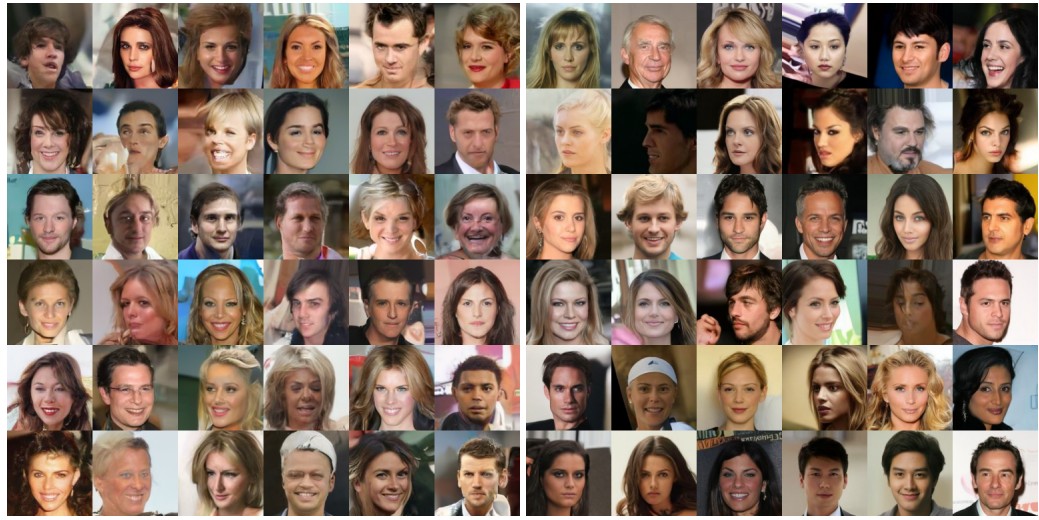

Figure 9: CELEB-A-HQ (128x128) samples. **Left**, the samples from DMMD, **Right**, the samples from DDPM.

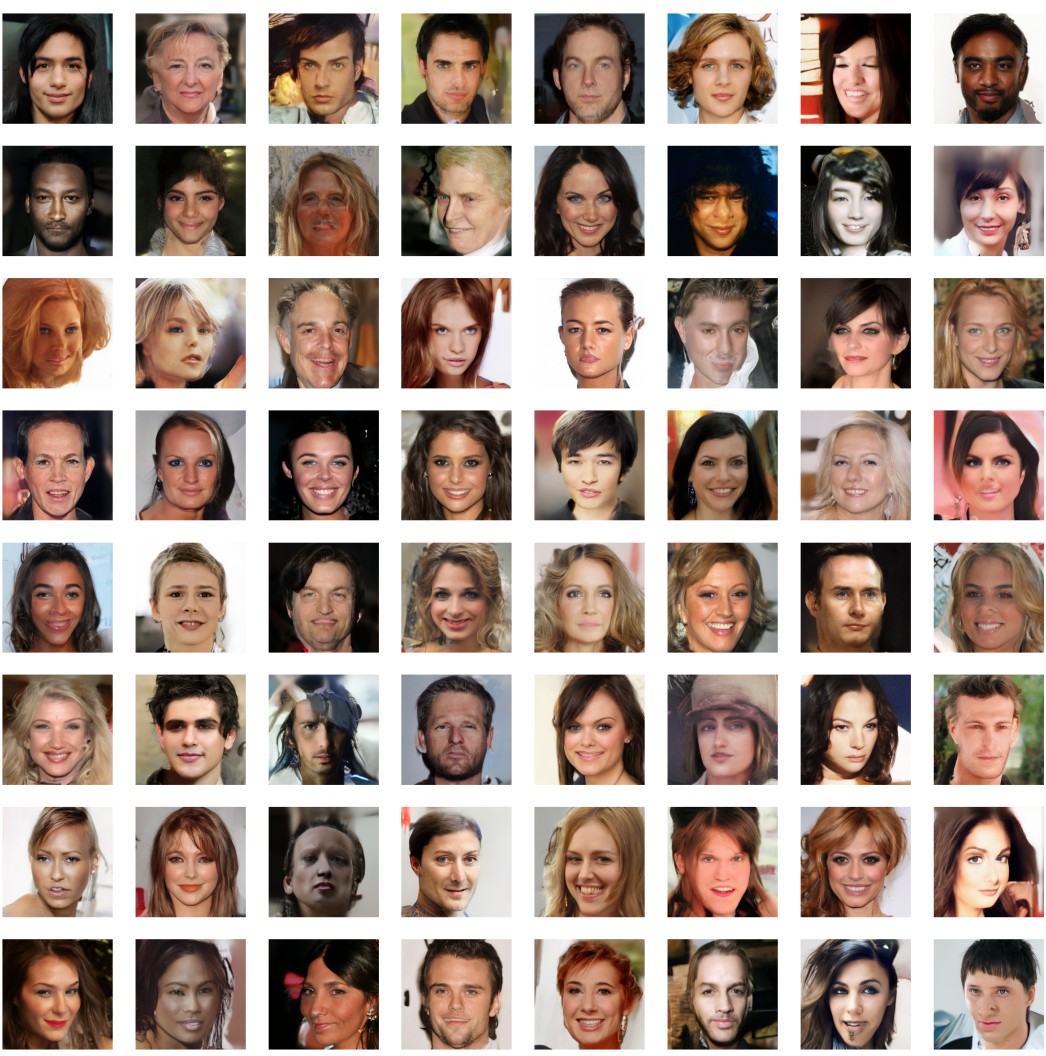

Figure 10: Latent (64x64x3) CELEB-A-HQ (256x256x3) samples for latent DMMD.

### L.3 ADDITIONAL DATASETS SAMPLES ON CELEB-A-HQ

The samples for CELEB-A-HQ (128x128) are given in Figure 9.

### L.4 ADDITIONAL DATASETS SAMPLES ON LATENT (64x64x3) CELEB-A-HQ (256x256x3)

The samples for latent DMMD are given in Figure 10 and the samples for latent diffusion are given in Figure 11.

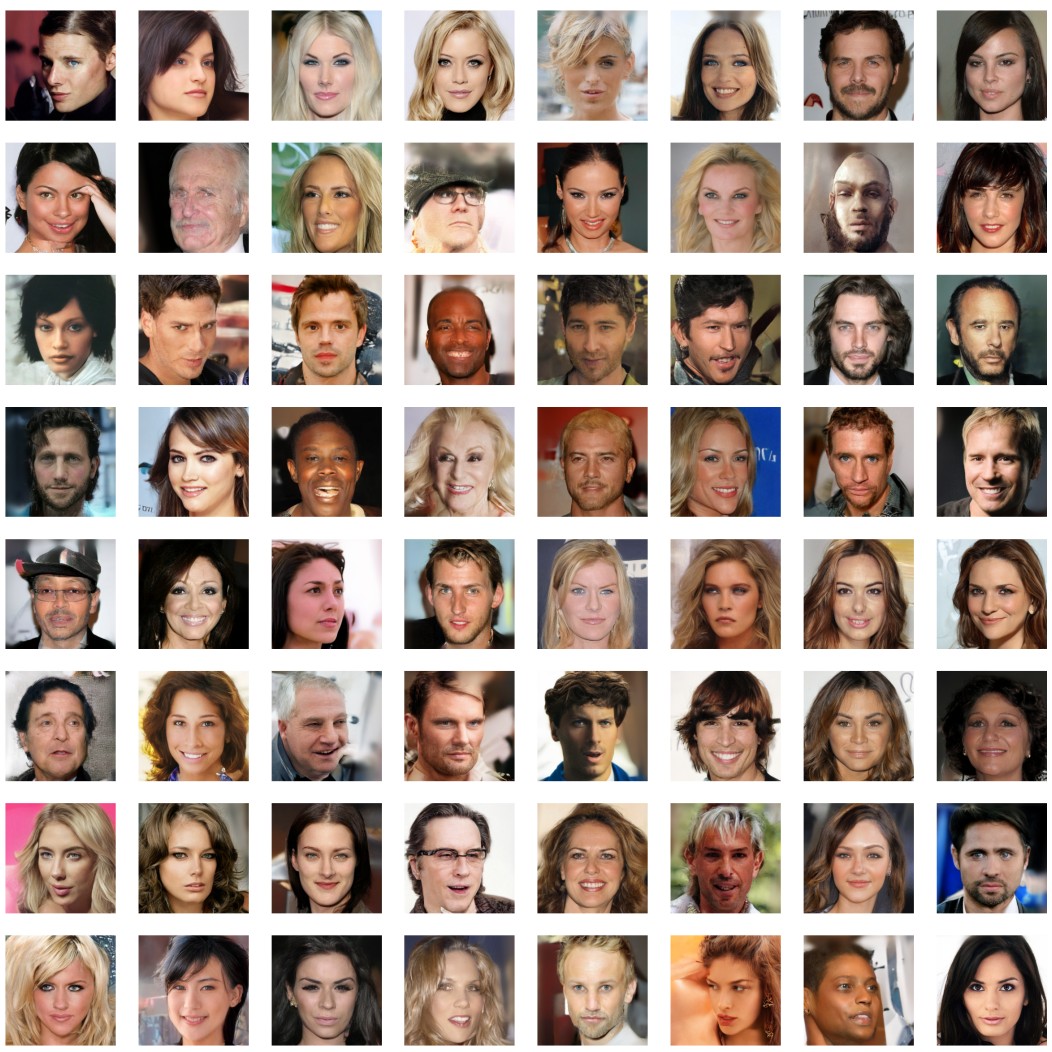

Figure 11: Latent (64x64x3) CELEB-A-HQ (256x256x3) samples for latent diffusion.

