## A  ORGANIZATION OF THE SUPPLEMENTARY MATERIAL

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

Using Proposition H.3, Proposition H.4 and definition (23), we have the following result.

**Proposition H.5.** *For any $\alpha, \sigma > 0$ and $\mu_0 \in \mathbb{R}^d$ we have*

$$\mathrm{MMD}^2(\pi_{0,\sigma}, \pi_{\mu_0,\sigma}) = 2(\alpha^2 + 2\sigma^2)^{-d/2}(1 - \exp[-\|\mu_0\|^2/(2(\alpha^2 + 2\sigma^2))]).$$

*In addition, we have*

$$\nabla_{\mu_0}\mathrm{MMD}^2(\pi_{0,\sigma}, \pi_{\mu_0,\sigma}) = -2(\alpha^2 + 2\sigma^2)^{-d/2-1} \exp[-\|\mu_0\|^2/(2(\alpha^2 + 2\sigma^2))]\mu_0.$$

Finally, we have the following proposition.

**Proposition H.6.** *For any $\mu_0 \in \mathbb{R}^d$ and $\sigma > 0$ let $\alpha^\star$ be given by*

$$\alpha^\star = \mathrm{argmax}_{\alpha \geq 0}\|\nabla_{\mu_0}\mathrm{MMD}^2(\pi_{0,\sigma}, \pi_{\mu_0,\sigma})\|.$$

*Then, we have that*

$$\alpha^\star = \mathrm{ReLU}(\|\mu_0\|^2/(d+2) - 2\sigma^2)^{1/2}.$$

*Proof.* Let $\sigma > 0$ and $\mu_0 \in \mathbb{R}^d$. First, using Proposition H.5, we have that for

$$\|\nabla_{\mu_0}\mathrm{MMD}^2(\pi_{0,\sigma}, \pi_{\mu_0,\sigma})\|^2 = 4\alpha^{2d}\|\mu_0\|^2(\alpha^2 + 2\sigma^2)^{-d-2} \exp[-\|\mu_0\|^2/(\alpha^2 + 2\sigma^2)].$$

Next, we study the function $f : [0, t_0] \to \mathbb{R}$ given for any $t \in [0, t_0]$ by

$$f(t) = t^{d+2} \exp[-t\|\mu_0\|^2],$$

with $t_0 = 1/(2\sigma^2)$. We have that

$$f'(t) = t^{d+1} \exp[-t\|\mu_0\|^2]((d+2) - \|\mu_0\|^2 t).$$

We then consider two cases. First, if $t_0 \leq (d+2)/\|\mu_0\|^2$, i.e. $\sigma^2 \leq \|\mu_0\|^2/(2(d+2))$, then $f$ is increasing on $[0, t_0]$ and we have that $f$ is maximum if $t = t_0$. Hence, if $\sigma^2 \leq \|\mu_0\|^2/(2(d+2))$, we have that $\alpha^\star = 0$. Second, if $t_0 \leq (d+2)/\|\mu_0\|^2$, i.e. $\sigma^2 \leq \|\mu_0\|^2/(2(d+2))$ then $f$ is increasing on $[0, t^\star]$, non-increasing on $[t^\star, t_0]$ with $t^\star = (d+2)/\|\mu_0\|^2$ and we have that $f$ is maximum if $t = t^\star$. Hence, if $\sigma^2 \geq \|\mu_0\|^2/(2(d+2))$, we have that $\alpha^\star = (\|\mu_0\|^2/(d+2) - 2\sigma^2)^{1/2}$, which concludes the proof. $\qquad\square$

### H.1 PHASE TRANSITION BEHAVIOUR

## I IMAGE GENERATION SAMPLES

### I.1 CIFAR10 SAMPLES

Samples from DMMD with NFE=100 and NFE=250 are given in Figure 4. Samples from DMMD with NFE=100 and from $a$-DMMD with NFE=50 are given in Figure 5.

### I.2 ADDITIONAL DATASETS SAMPLES

Samples for MNIST are given in Figure 6, for CELEB-A (64x64) are given in Figure 7 and for LSUN Church (64x64) are given in Figure 8.

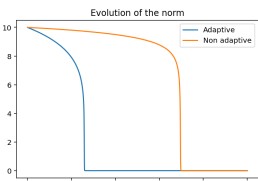 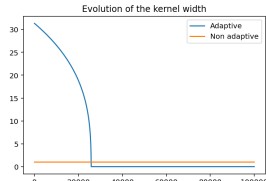

Figure 3: Evolution of the norm of the mean $\mu_t$ of the Gaussian distribution $\pi_{\mu_t,\sigma}$ according to a gradient flow on the mean $\mu_t$ w.r.t. $\mathrm{MMD}_{\alpha_t}$. In the *adaptive* case $\alpha_t$ is given by Proposition 3.1 while in the *non adaptive* case, $\alpha_t = \alpha_0 = 1$. In our experiment we consider $d = 1$ and $\sigma = 1$, for illustration purposes.

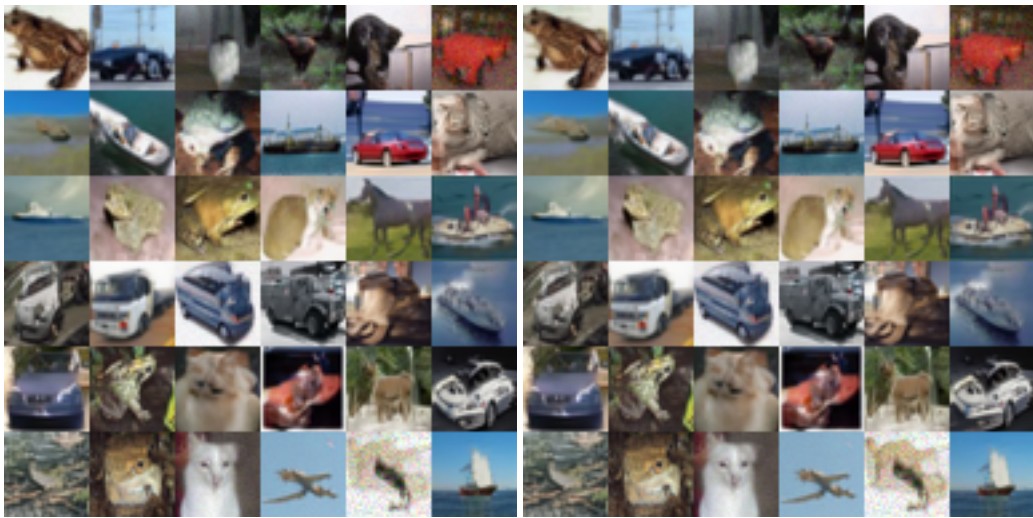

Figure 4: CIFAR-10 samples from DMMD with NFE=250 on the left and with NFE=100 on the right

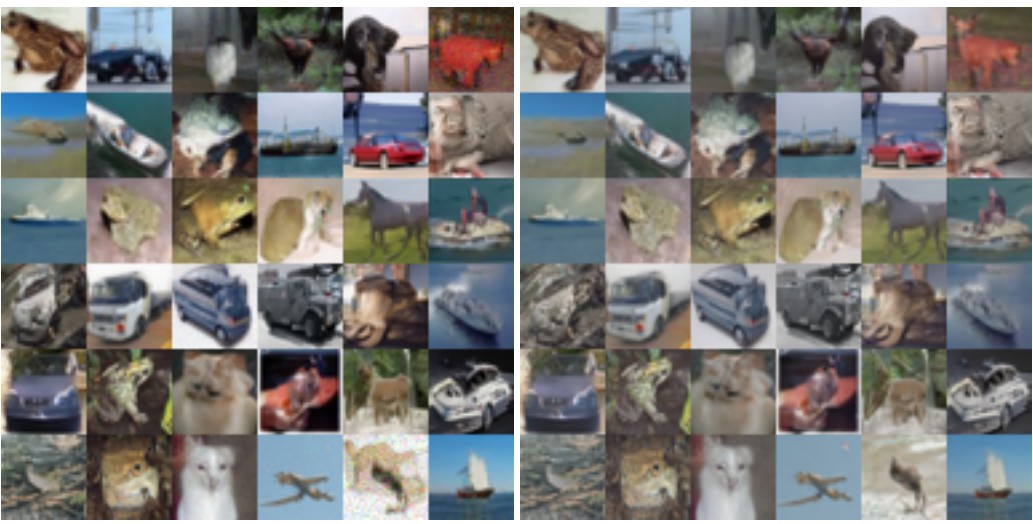

Figure 5: CIFAR-10 samples from DMMD with NFE=100 on the left and samples from the $a$-DMMD-$e$ with NFE=50 on the right

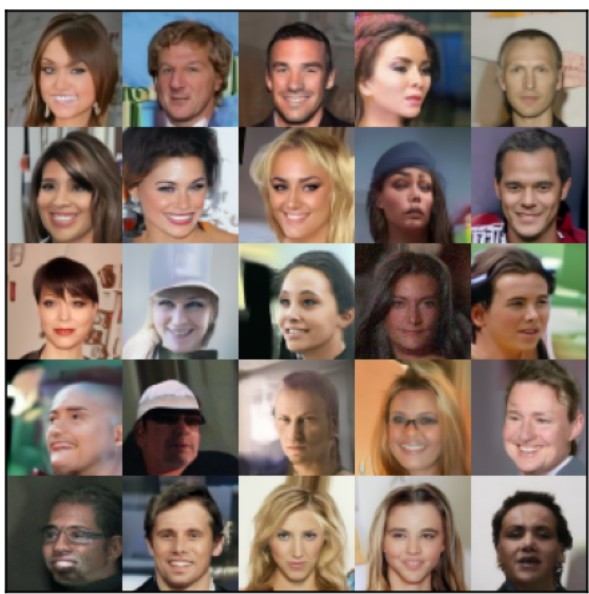

Figure 6: DMMD samples for MNIST.

Figure 7: DMMD samples for CELEB-A (64x64).

1512
1513
1514
1515
1516
1517
1518
1519
1520
1521
1522
1523
1524
1525
1526
1527
1528
1529
1530
1531
1532
1533
1534
1535
1536
1537
1538
1539
1540
1541
1542
1543
1544
1545
1546
1547
1548
1549
1550
1551
1552
1553
1554
1555
1556
1557
1558
1559
1560
1561
1562
1563
1564
1565

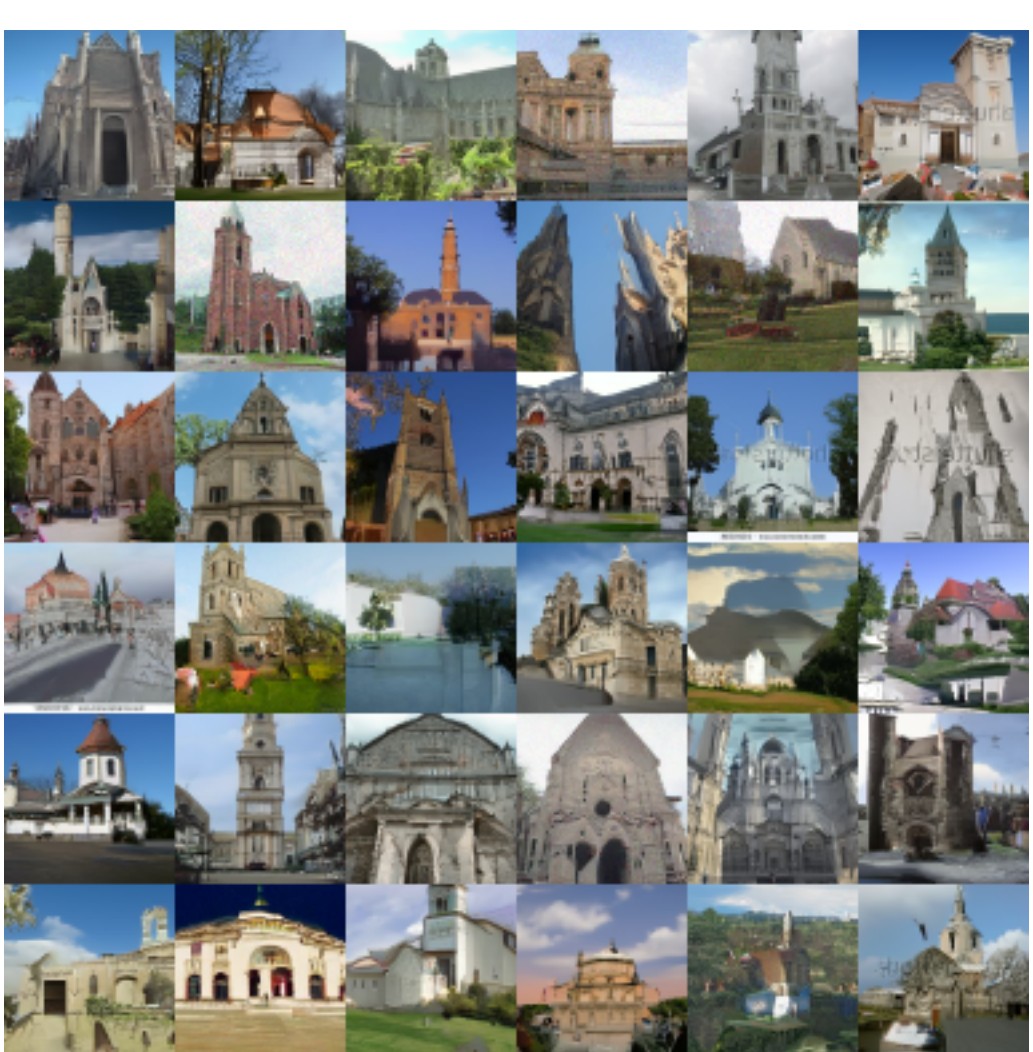

Figure 8: DMMD samples for LSUN Church (64x64).