# OpenReview forum: "Deep MMD Gradient Flow without adversarial training"
_ICLR.cc/2025/Conference — ICLR 2025 Poster_

### Official Review · Reviewer_vKuM · 2024-10-27

**Soundness:** 2
**Presentation:** 3
**Contribution:** 2
**Rating:** 6
**Confidence:** 3

**Summary:**

This paper proposes a generative model based on the Wasserstein gradient flow of the Maximum Mean Discrepancy (MMD), called DMMD. Given the forward process in diffusion models, the proposed method learns a discriminator between clean data and noisy data. The Wasserstein gradient flow is represented via this discriminator. Therefore, DMMD generates samples by simulating particle trajectories through this flow. DMMD achieves superior performance compared to other discriminator flow and MMD flow baselines in image generation across various datasets.

**Strengths:**

- This paper is easy to follow.
- This paper suggests an efficient approximate sampling procedure for the linear kernel.
- DMMD is evaluated on various image datasets, such as CIFAR-10, MNIST, CelebA (64x64), and LSUN-Church (64x64).

**Weaknesses:**

- My main concern is whether the trajectories of the probability distributions for the forward process $\\{ p^{1}\_{t} \\}\_{t \geq 0}$ in the diffusion model and the MMD gradient flow $\\{ p^{2}\_{t} \\}\_{t \geq 0}$ coincide. If these trajectories are different, DMMD learns the Wasserstein gradient flow for minimizing $MMD (p^{1}\_{t}, p_{data})$. However, during the generation process, the particles follow $\\{ p^{2}\_{t} \\}$ at $ t - \triangle t$. This gradient flow mismatch cannot guarantee that the particles correctly generate the target distribution. Could you clarify this concern?
- The generation process is computationally expensive. How many $N_{s}$ steps are required for each time $t$ during sample generation (Eq 9)?
- For a general kernel, the sampling from DMMD requires access to the training data (Eq 10). Only for linear kernel, this issue can be avoided by saving the average features for each time $t$ (Eq. 12).

**Questions:**

- In Table 1, do all the other baselines use the same backbone network?
- Could you provide the wall-clock time comparison to demonstrate the computational benefits of approximate sampling in Table 2?
- Is this method applicable to higher-dimensional datasets, such as CelebA-HQ (256x256)?
- Could you provide the potential advantages of this approach compared to other non-adversarial dynamic generative models, such as Flow Matching [1]?

[1] Lipman, Yaron, et al. "Flow matching for generative modeling." ICLR 2023.

---

> ### Author Response · Authors · 2024-11-21
> **Answer to reviewer vKuM**
>
> We thank the reviewer vKuM for their feedback. Please find our detailed answer below.
>
> > ...trajectories of the probability distributions for the forward process in the diffusion model and the MMD gradient flow coincide ?
>
> This is indeed an important point. The trajectories might not coincide because in principle if one starts with particles $X_t|X_0$, one step of MMD gradient flow with $MMD_t$ (MMD which uses features conditioned on t), might produce particles Z which do not correspond to a noise level t’=t-dt, seen during training. Even if the noise levels do not coincide with the forward process, however, MMD_{t} is a valid MMD for any t in $[0,1]$ implying that we can take its corresponding MMD gradient flow. Following these gradient flows would bring us closer to the target distribution $P_0$ and the only potential problem is the speed of convergence.
>
> Since $t$ corresponds to a noise level from the forward diffusion process, $MMD_{t}$ for $t\sim1$ was trained to distinguish between clean data and noise and $MMD_{t}$ for $t\sim0$ was trained to distinguish between clean data and slightly noisy data. Making the parallel with a Gaussian 2D example (see Figure 2), $t$ would encode the kernel scale parameter. If one follows the MMD gradient flow when $t$ is large, the gradient is only large if the samples are very noisy and is small otherwise. In order to avoid the case where the sampled noisy particles have noise level $t'$ higher than $t$ used for $MMD_{t}$, we use a two stage procedure described in Algorithm 2, where for each sampling-time noise level, we do multiple MMD gradient flow steps. This allows us to gradually decrease noise level to keep MMD gradient high.
>
> > The generation process is computationally expensive. How many steps are required for each time during sample generation (Eq 9)?
>
> Please see Appendix F for more details. For most of the experiments, we use $100$ steps in total which do $N_{s}=10$ gradient flow steps for each of $N_{noise}=10$ noise levels.
>
> > For a general kernel, the sampling from DMMD requires access to the training data (Eq 10). Only for linear kernel, this issue can be avoided by saving the average features for each time (Eq. 12).
>
> That's correct. But since we use a linear kernel on top of the learned Neural Network features, the overall resulting kernel is in fact not a linear kernel but rather a flexible learned kernel, see eq.15 in Appendix B. In fact, we explored using RBF and Rational Quadratic kernels instead of linear, for image generation and we did not find that it leads to the improvement in performance.
>
> > Could you provide the wall-clock time comparison to demonstrate the computational benefits of approximate sampling in Table 2?
>
> Please see our answer above the computational complexity. As we state above, during sampling time, if we use linear kernels with ordinary witness function, the computational complexity of one sampling step is $O(N)$, where $N$ is the number of samples which we produce simultaneously (due to the interacting particles structure). When using approximate sampling, we decrease this complexity to $O(1)$.
>
> > Is this method applicable to higher-dimensional datasets, such as CelebA-HQ (256x256)?
>
> Please see our common answer to all the reviewers.
>
> > Could you provide the potential advantages of this approach compared to other non-adversarial dynamic generative models, such as Flow Matching [1]?
>
> Diffusion models try to reverse a Gaussian noising dynamic while flow matching methods try to match an interpolation dynamics between the data distribution and a Gaussian distribution. Both methods can be shown to be equivalent given the choice of the schedule. In particular, both methods are relying on the fact that the target distribution is Gaussian and that the forward process is given by a Ornstein-Uhlenbeck (or Brownian motion) SDE.
>
> On the other hand, our approach is agnostic to the destruction process and does not rely on an underlying Gaussian assumption of the prior distribution. In particular, one could design destruction processes which are more adapted to the data. While extensions have been proposed for such destruction processes (see [1,2] for instance) they either rely on a soft Gaussian assumption for the noising process [2] or do not enjoy the theoretical properties of our framework [1], i.e. at optimality we recover the data distribution, up to the discretization error.
>
> We will add this discussion on the relation to flow matching to the final version.
>
> **References**:
>
> [1] Bansal et al. – Cold Diffusion: Inverting Arbitrary Image Transforms Without Noise
>
> [2] Daras et al. – Soft Diffusion: Score Matching for General Corruptions

---

> > ### Comment · Reviewer_vKuM · 2024-11-24
> >
> > Thank you for the response. However, my main concern (W1) remains unaddressed. Specifically, the problem is that we cannot guarantee that the generated particles $p_{t}^{2}$ at $t-\Delta t$ are the Gaussian convolution of the data distribution. For example, consider a scenario where we are given a velocity field $v(x)$ that drives every $x$ to converge to a stationary point $x^{*}$. Under the current method, the particles at $x_{1}$ would be updated using the velocity field $v(x_{2})$ where $x_1 \neq x_2$. This part raises important concerns about the soundness of the proposed method. One possible way to address this concern would be to provide an upper bound on the distance between $p_{t}^{1}$ and $p_{t}^{2}$.
> >
> > Moreover, the Flow Matching does not require the Gaussian assumption for the prior distribution. This model only requires some joint distribution between the input and output distributions, such as independent coupling or optimal transport plan.
> >
> > For these reasons, I would keep my original rating.

---

> > > ### Author Response · Authors · 2024-11-26
> > > **Answer**
> > >
> > > > (W1)
> > >
> > > Thank you for getting back to us quickly - this is an important question, and we agree we weren’t as clear as we should have been in our last response. We will give a more detailed response below, and attempt to improve on our earlier explanation.
> > >
> > > First of all, to set notation: at every time t, we maximize the $MMD_t[X_0; X^1_t; t]$ (with regularization, see eq. 8) on the forward diffusion $P_t^1$. We should consider carefully what this means: namely, that we find the best kernel features $\phi(x; t)$ at temperature t, in order to distinguish samples at $X^1_t$ from those at $X_0$.
> > >
> > > Now: what happens if we use the kernel features $\phi(x; t)$ to define the gradient of an MMD flow, on a sample $X^2_s$ from the backwards flow for some temperature s (which needn’t be t). **This is still a valid MMD**, i.e. a difference in expected feature means! What’s more, it is **still a valid divergence** even if the current backwards $X^2_t$ is not exactly $X_0$ convolved with a Gaussian of the bandwidth corresponding to t. This means that it will still provide a gradient field on the sample $X^2_s$ that points to the target $X_0$.
> > >
> > > There’s nonetheless a good reason to choose features $\phi(x; t)$ for the MMD flow on $X^2_t$ that (roughly) match the temperature t of corresponding samples $X^1_t$ in the forwards diffusion process. This is because MMD_s for $s \neq t$ won’t provide as fast a convergence as MMD_t (but it *will* still point the samples $X^2_t$ in the right direction, since it is still a valid divergence measure). We achieve this with a conservative heuristic, which is to take $N_s$ steps of MMD flow per noise level, before reducing the temperature (Algorithm 2). This seems to work well in practice, in that the particle population isn’t left with “stragglers” that haven’t followed the gradient field.
> > >
> > > We’d welcome any further questions on the above, and hope the explanation makes more sense.
> > >
> > > As an aside, we emphasize that in practice, **score-based generative networks have the same limitation**. Namely, since the score is learned, it could also happen that the samples at time $t$ from the backward process, do not follow the forward diffusion process with noise level $t$. In practice, and as with MMD flow, this is not generally an issue.
> > >
> > > It is also common in diffusion models to consider predictor-corrector schemes, see [1]. In that case, sampling at a noise level t is performed for a few iterations (corrector) before performing the jump to another noise level (predictor). Our scheme can be seen as a “corrector only” scheme in the framework of [1], where instead of using the Langevin algorithm, we follow MMD flows.
> > >
> > > We are happy to add all these points, including the link to predictor-corrector schemes,  to a revised version of the paper.
> > >
> > > **References**:
> > >
> > > [1]  Song – Score-Based Generative Modeling through Stochastic Differential Equations (2021)
> > >
> > >
> > > > Moreover, the Flow Matching does not require the Gaussian assumption for the prior distribution. This model only requires some joint distribution between the input and output distributions, such as independent coupling or optimal transport plan.
> > >
> > > It is true that the Flow Matching formulation can get rid of the Gaussian prior assumption. However, we highlight that the original paper “Flow Matching” focuses solely on the Gaussian case (generalizations were introduced in similar works) such as Stochastic Interpolant or Inversion by Direct Iteration.
> > >
> > > If we consider the general case where the interpolation is given by a stochastic interpolant (see [1], Equation 2.1), where we denote $I(t, x_0, x_1)$ the interpolant, then we need to have access to the derivative with respect to $t$ of this interpolant in order to learn the velocity function. In the case of DMMD, we do not make any requirements on $I(t, x_0, x_1)$ other than to be able to compute it (no differentiability assumption, no need to access its differential etc).
> > >
> > > Hence, DMMD allows more general perturbation than the ones considered in stochastic interpolants. This is what we meant by “our approach is agnostic to the destruction process”.
> > >
> > >
> > > **References**:
> > >
> > > [1] Albergo et al., Stochastic Interpolants:A Unifying Framework for Flows and Diffusions

---

> > > > ### Comment · Reviewer_vKuM · 2024-11-30
> > > >
> > > > Thank you for the clarifications regarding the validity of MMD. These have been helpful in addressing my concerns. Therefore, I will increase my rating to 6.

---

### Official Review · Reviewer_xQpY · 2024-11-02

**Soundness:** 3
**Presentation:** 3
**Contribution:** 2
**Rating:** 6
**Confidence:** 3

**Summary:**

The article proposes a gradient flow procedure for generative modeling by transporting particles from an initial source distribution to a target distribution, where the gradient field on the particles is given by a noise-adaptive Maximum Mean Discrepancy (MMD) divergence. The noise adaptive MMD is trained on a data distribution corrupted by forward diffusion process. The divergence training procedure is related to discriminator training in Generative Adversarial Networks (GAN), but does not require adversarial training. The article demonstrates competitive empirical performance of the method in unconditional image generation on CIFAR10 dataset.

**Strengths:**

The article explores the integration of a diffusion process into models based on Maximum Mean Discrepancy (MMD) gradient flow, offering a theoretical foundation for both the discriminator and the sampling process used in MMD-GAN with this diffusion approach. Additionally, it employs a linear kernel for the scalable MMD-GAN to reduce computational complexity. The authors also conduct experiments using other forms of KL divergence, such as KALE divergence, to demonstrate its effectiveness.

**Weaknesses:**

1. I believe the contribution of this article is inadequate. Previous research has utilized the diffusion process in the discriminator, as noted in this work [1]. However, this article does not provide theoretical proof demonstrating that the MMD GAN can converge to more optimal points when using the diffusion process.

2. Additionally, the effectiveness of MMD Gradient Flow has only been tested on low-resolution datasets, which does not provide sufficient evidence to confirm its overall efficacy. I recommend that the author conduct additional experiments using high-resolution datasets at a resolution of 256x256, specifically on the LSUN and CelebA datasets. These experiments should include evaluations based on the number of metric sampling steps (NFE) and diversity (FID).

[1] Xiao Z, Kreis K, Vahdat A. Tackling the generative learning trilemma with denoising diffusion

**Questions:**

1. I recommend that the author conduct additional experiments using a variety of datasets to demonstrate the effectiveness of the DMMD. The results on CIFAR10 are not as impressive as those achieved by state-of-the-art generative models. I also suggest that the author include more comparative experiments with additional relevant works.

2. I am wondering if using vector Z in Eq.(15) to train a generator will result in better outcomes than using it as a sampling process.

3. I am curious about the time required to train the MMD Gradient Flow on the CIFAR-10 dataset. In my experience, training a diffusion-based discriminator takes significantly longer than training the original models. I find that incorporating a diffusion process in gradient flow-based models can reduce the number of accumulated gradient steps. I would like to know if the sampling steps of the DMMD model are influenced by the diffusion process.

4. I wonder how the sampling time changes with increasing data dimensions, such as in 256x256 high-dimensional datasets.

5. Can the MMD Gradient Flow be applied to other tasks such as Super-Resolution or Inpainting? These tasks also involve particle transportation from an initial source to a target distribution.

6. There are writing errors in the article, particularly in Table 4 and Algorithm 2.

---

> ### Author Response · Authors · 2024-11-21
> **Answer to Reviewer xQpY, Part 1**
>
> We thank reviewer xQpY for the feedback and suggestions. Please find our answers below. We would be happy to further discuss any additional questions that might arise. If we have satisfactorily addressed your questions, we would be grateful if you could reconsider your score.
>
> > prev. research utilized diffusion process in the discriminator [1]
>
> Thank you for pointing out this connection. We would like to highlight that eq.(6) in our paper corresponds to the forward diffusion (FD) process and eq. (6) in [1] also corresponds to the FD process specified in their appendix. If your point was that we both use a FD process, then it is a correct observation.
>
> Please note, however, that even though at first glance, the discriminator objective in [1] and in our paper look similar, they are in fact quite different, since in [1], the authors construct a discriminator to distinguish between $x_{t+1}$ produced by a trained generator (see eq.5 in [1] and eq.4 in [1]) and $x_{t}$ produced by the forward diffusion process, for a given noise level $t$. This difference arises because the purpose and algorithm in [1] are fundamentally different to ours: the authors of [1] aim to train a GAN conditional generator which can do big denoising steps (vs the small, Gaussian denoising steps in a traditional diffusion model).
> Our objective function is $D[x0; xt; t]$, where $x_0$ is clean data and $x_t$ is noisy data from the forward diffusion process, and $D$ is the discriminator divergence. Our objective function aims to build a discriminator that is able to distinguish clean data $x_0$ from noisy data $x_t$ produced by the forward diffusion with a given level of noise $t$. Here the noise level $t$ is roughly  (see Section 4 in our paper for discussion as well as experiments on toy data) related to the quality of the GAN generator (if we were to use a GAN generator), such that $t\rightarrow1$ corresponds to a generator at the beginning of the training and $t\rightarrow0$ corresponds to a generator at the end of the training. We emphasize that this is only an intuitive correspondence, since we don’t actually use a generator!
>
> Therefore, the optimization problem in [1] is adversarial since it involves training generator/discriminator pairs (so as to take large denoising steps), whereas the optimization problem in our work is not adversarial since we use a fixed sequence of distributions (from the forward diffusion process) to train a discriminator across corresponding noise levels. See our reply above for more details on adversarial training.
>
> > ...no theoretical proof for MMD GAN can converge to more optimal points with diffusion process
>
> If we have understood this question correctly, it addresses whether an MMD diffusion (with no generator) will be able to better match the target distribution than an MMD GAN (with a generator). This will be the case if the generator has insufficient capacity to represent the target distribution, compared with a collection of particles. For instance, if the generator is a single Gaussian, but the target is a mixture of two Gaussians, then the MMD diffusion can attain the target distribution, but the MMD GAN will not. It appears from empirical studies that limitations in generator capacity are one major reason why diffusions have now taken over as the dominant image generating paradigm (the other is that adversarial training is difficult due to its min-max nature). In all experiments on images, the MMD flow strongly outperforms the MMD GAN.
>
> > Low resolution and SOTA
>
> Please see the answer to this point in the common answer section.
>
> >training time...
>
> See our common answer above regarding computational complexity of DMMD.
>
> >impact of forward process...
>
> We have not investigated the impact of the diffusion process on the DMMD model performance but took the default forward process from the DDPM paper [1]. As we explain in our common answer, investigating the impact of forward process might improve DMMD performance and is the subject of future work.
>
> >sampling time dependent on resolution?
>
> At sampling time, we compute gradient of MMD witness function which is 2x cost of the forward pass, and the forward pass cost scales proportionally to the resolution.
>
> >There are writing errors in the article, particularly in Table 4 and Algorithm 2.
>
> Thank you for bringing these to our attention. We will remedy these errors in a final version.
>
> **References**:
>
> [1] Denoising Diffusion Probabilistic Models, Jonathan Ho, Ajay Jain, Pieter Abbeel
>
> [2] Johnson R, Zhang T. Composite functional gradient learning of generative adversarial models[C]//International Conference on Machine Learning. PMLR, 2018: 2371-2379.
>
> [3] Unifying GANs and Score-Based Diffusion as Generative Particle Models Jean-Yves Franceschi, Mike Gartrell, Ludovic Dos Santos, Thibaut Issenhuth, Emmanuel de Bézenac, Mickaël Chen, Alain Rakotomamonjy, 2023
>
> [4] Deep Generative Wasserstein Gradient Flows, Alvin Heng, Abdul Fatir Ansari, Harold Soh, 2023

---

> ### Author Response · Authors · 2024-11-21
> **Answer Part 2 -- Adversarial training clarification**
>
> High-level clarifications about adversarial training
> In anticipation of follow-up questions that might arise, we begin by clarifying what we mean by adversarial training. We will add this discussion in the Appendix of the revised version of our paper.
>
> In the context of Generative Adversarial Networks (GANs), the objective function is
>
> $F(\theta, \psi) = \mathbb{E}_{Z \sim U[-1,1]} D[X; G(Z; \psi); \theta]$,
>
> where $G(Z, \psi)$ is a generator with parameters $\psi$ and $D[X; G(Z; \psi); \theta]$ is a discriminator divergence with parameters $\theta$.
>
> Objective for the generator:
> $\psi^* \leftarrow \arg\min_{\psi}\max_{\theta} F(\theta, \psi)$
>
> Objective for the discriminator:
> $\theta^* \leftarrow \arg\max_{\theta}\min_{\psi} F(\theta, \psi)$
> In practice in GANs, this objective is approximated by alternating updates. Let $n$ be the iteration number.
>
> Update generator as:
> $\psi^{n+1} \leftarrow \arg\min_{\psi} F(\theta^n, \psi)$
>
> Update discriminator as:
> $\theta^{n+1} \leftarrow \arg\max_{\theta} F(\theta, \psi^{n+1})$

---

> ### Author Response · Authors · 2024-11-21
> **Answer Part 3**
>
> >Inpainting & Super-resolution?
>
> In principle, one could use DMMD for tasks which go beyond generative modeling. In this work, we did not investigate the performance of DMMD on such tasks. We believe that in order to properly test DMMD in these scenarios, more work is required since it is typically not the case that using a model in these scenarios works out of the box. We appreciate your suggestion and think it would be worthwhile to study these use-cases in future work.
>
> > vector Z in Eq.(15) to train a generator...
>
> We believe you refer to eq. (9) and not to eq.(15), since eq.15 refers to a kernel $k(x,y)$ in Appendix B. It is not entirely clear to us what you mean by “using vector Z in Eq.(15) to train a generator”. What generator are you referring to?
>
> We do not have a generator in our method. We use noise-adaptive MMD gradient flow only at sampling time which essentially replaces a generator. At training time, we only use a fixed forward diffusion process. It is unclear to us what you mean when you mention using vector Z from eq.15 to train a generator, but we would be happy to discuss further if you can provide further details on this idea.
>
> In anticipation of future questions that might arise: you might be thinking of the fact that Eq.3, Eq.4, Eq.,9 in the MMD flow are the same theoretical form as the functional gradient flow Eq.(5) in the article [2], where eq.5 is used to train a generator to approximate the sample from the target distribution. Moreover, in our paper, eq.9 employs a negative gradient from the discriminator, so you might further suppose that this would result in an adversarial process – something which is used in [2]. This is, however, not correct because there is a crucial difference between our work and [2].
>
> In [2], there is the alternation of training of a generator and a discriminator as with Generative Adversarial Networks (GAN): see our point about adversarial training. The generator is updated at each step using a "functional" update, which involves taking the previous generator at time $t-1$ and updating it in the optimal direction indicated by the current discriminator at time $t$ (in the case of KL, the update direction is the derivative of the log density ratio given by the discriminator): see their Algorithm 1 Step 4. This means that their generator is expressed as a weighted sum of derivatives of the discriminator networks at previous steps: see their Alg. 1 Step 4.
>
> There is a fundamental and critical difference between [2] and our method: in [2], a discriminator is trained at each step on particles produced by the generator at the previous step. This means that the procedure is adversarial: the generator moves in the direction that minimizes the discriminator loss; the discriminator then gets updated to distinguish current generator samples from real samples; and the cycle repeats. By contrast, our method is not trained adversarially. The training samples are provided by a forward diffusion, which is fixed in advance, and never changes, and is obtained simply by adding increasing levels of noise to clean samples. This is a fundamental difference with [2], where the training samples are the output of a generator, which evolves as the discriminator evolves. There is no generator in our approach.
>
> The approach of [2] is in fact quite related to the approaches [3-4] cited in our paper, which accomplish similar adversarial updates (where this process is called "discriminator flow"). A challenge in using a discriminator flow approach, also noted in both [2] and in our paper (line 301), is that it requires maintaining a large number of intermediate discriminator networks to define the final generator at the end of the training process (see figure 1 in [2]), resulting in a generator of large size and high computational cost. Heuristics may be used to approximate this large network (as in [2]). Our approach does not suffer from this high computational overhead.
>
> We think it's worthwhile for us to cite [2] in a revised version of our paper, and to add discussion of [2] to the literature review.
>
> **References**:
>
> [1] Denoising Diffusion Probabilistic Models, Jonathan Ho, Ajay Jain, Pieter Abbeel
>
> [2] Johnson R, Zhang T. Composite functional gradient learning of generative adversarial models[C]//International Conference on Machine Learning. PMLR, 2018: 2371-2379.
>
> [3] Unifying GANs and Score-Based Diffusion as Generative Particle Models Jean-Yves Franceschi, Mike Gartrell, Ludovic Dos Santos, Thibaut Issenhuth, Emmanuel de Bézenac, Mickaël Chen, Alain Rakotomamonjy, 2023
>
> [4] Deep Generative Wasserstein Gradient Flows, Alvin Heng, Abdul Fatir Ansari, Harold Soh, 2023

---

> > ### Comment · Reviewer_xQpY · 2024-11-26
> > **Response to author**
> >
> > Thank you for your response. I appreciate the authors' great effort in the Wasserstein Gradient Flow. Although we have differing views on adversarial learning, the MMD gradient flow is complex and does not yield particularly desirable effects; however, it still provides some useful results for gradient-flow-based methods. I believe that exploring the use of gradient flow in generative models is worthwhile, so I'm willing to increase my rating.

---

### Official Review · Reviewer_KT31 · 2024-11-04

**Soundness:** 3
**Presentation:** 3
**Contribution:** 3
**Rating:** 5
**Confidence:** 2

**Summary:**

The paper integrates MMD GAN with the diffusion forward process, introducing a novel generative model framework called DMMD. Through the design of a noise-adaptive MMD gradient flow, this framework aims to reduce the challenges of adversarial training and address the singularity issues found in score-based methods.

**Strengths:**

S1 - The paper provides a well-formulated background and problem statement, with a theoretically motivated and well-grounded DMMD framework.

S2 - The idea of using an adversarial training-free discriminator based on the diffusion forward process could offer valuable insights to the community.

**Weaknesses:**

W1 - The framework's absolute performance is a concern, as DMMD shows a significant performance gap compared to DDPM and more modern methods on the selected image generation benchmarks.

W2 - Its broader application potential is limited, with empirical evaluation restricted to small datasets like MNIST and CIFAR.

W3 - The sampling method appears restrictive, requiring reference features from the ground truth dataset to formulate the witness function.

**Questions:**

Q1 - The need for dataset features during sampling seems counterintuitive. Do the authors have any insights into potential solutions for addressing this limitation?

---

> ### Author Response · Authors · 2024-11-21
> **Answer to Reviewer KT31**
>
> We thank reviewer KT31 for the feedback and suggestions. Please find our responses below. We would be happy to further discuss any additional questions that might arise. If we have satisfactorily addressed your questions, we would be grateful if you would consider increasing your score.
>
> > The framework's absolute performance is a concern, as DMMD shows a significant performance gap compared to DDPM and more modern methods on the selected image generation benchmarks.
>
> Please see our common response above regarding SOTA.
>
> > Its broader application potential is limited, with empirical evaluation restricted to small datasets like MNIST and CIFAR.
>
> Please see our common answer above.
>
> > The sampling method appears restrictive, requiring reference features from the ground truth dataset to formulate the witness function.
>
> The sampling method comes from the fact that the witness function is by construction explicitly dependent on the data distribution of the target, since the generated particles are required to interact with the target particles, as well as with each other. Any method which applies Wasserstein Gradient Flow to MMD metric will have a similar property in the sampling method.
>
> That being said, **one of the main contributions of this work is to provide a solution to this limitation**, and to remove this restriction, by pre-encoding the features of the target dataset into the average features (See Section 5, especially lines 313-315 and eq.12).
>
> > The need for dataset features during sampling seems counterintuitive. Do the authors have any insights into potential solutions for addressing this limitation?
>
> As mentioned above, our implementation does not require dataset features through the pre-encoding procedure. See the response to the previous comment.

---

> > ### Comment · Reviewer_KT31 · 2024-11-26
> >
> > Thank you for your response. I appreciate the authors' exploratory effort and the attempt to scale up to higher resolution despite the resource limitation. Despite the slight concern of Wasserstein Gradient Flow over Diffusion (higher training cost & requiring **access** to target dataset for sampling, pre-coding or not), I believe this work poses non-trivial improvement to Wasserstein Gradient Flow based methods and I'm thus willing to increase my rating.

---

> > > ### Author Response · Authors · 2024-11-27
> > > **Remark**
> > >
> > > > Thank you for your response. I appreciate the authors' exploratory effort and the attempt to scale up to higher resolution despite the resource limitation. Despite the slight concern of Wasserstein Gradient Flow over Diffusion (higher training cost & requiring access to target dataset for sampling, pre-coding or not), I believe this work poses non-trivial improvement to Wasserstein Gradient Flow based methods and I'm thus willing to increase my rating.
> > >
> > > Dear Reviewer KT31, we appreciate you taking time to reply to our answer, and we are happy to hear your assessment that the improvements over WGF methods are nontrivial.
> > >
> > > We would like to clarify one of your concerns with regards to the requirement of having access to the target dataset for sampling.
> > >
> > > We agree that if our method were used with a general kernel (eg Gaussian, inverse multiquadric), we would need to retain the target dataset to generate new samples.
> > >
> > > Our implementation takes a different approach, however: we use a linear kernel defined on learned neural net features. This allows an explicit representation of the gradient field as an average of the features of the target dataset,  $\bar\phi(X_0,t;\theta^*)$. See eq. 13, and the discussion form line 316 under the heading  “approximate sampling procedure.” What this means is that we do not require access to the target dataset, or  even to “pre-coded” samples from the target dataset, but only to the average of the target dataset features for each t. It is this average which is stored, and not the (encodings of the) individual target data.
> > >
> > > This brings our method into line with standard score-based diffusion models: in that case, the knowledge of the target data is “encoded” in the score network. In our case, it is encoded in the **mean feature vector** of the target.
> > >
> > > Please let us know if this is clear -  we’d be happy to answer any further questions.

---

> > > ### Author Response · Authors · 2024-12-01
> > > **follow up regarding target data**
> > >
> > > Dear Reviewer KT31, As the discussion process will end soon, we wanted to get in touch again regarding our previous reply.
> > > This relates to  your remaining concern (see remark below).  In our reply, we explain that we do not need access to the target dataset (in raw form or in pre-coded form) in order to generate samples. Only an average of features across all the target data is used - see reply for details.
> > >
> > > We would be happy to hear your feedback regarding the last point. If we have satisfied this concern, we would  be grateful if you would again consider raising your score.

---

### Official Review · Reviewer_QpwN · 2024-11-08

**Soundness:** 3
**Presentation:** 2
**Contribution:** 3
**Rating:** 6
**Confidence:** 2

**Summary:**

The paper presents a novel approach, "Diffusion Maximum Mean Discrepancy Gradient Flow" (DMMD), to improve generative modeling by combining Maximum Mean Discrepancy (MMD) with a noise-adaptive gradient flow mechanism. Unlike GANs, DMMD eliminates adversarial training by utilizing a noise-conditional MMD discriminator. DMMD introduces a particle transport technique, adapting MMD as the divergence metric to transport particles from a source distribution to a target distribution.

**Strengths:**

1. The paper is clearly written and well-organized, tackling a genuine problem and effectively presenting its contributions and findings.

2. The paper establishes a solid mathematical foundation, rigorously linking each section to prior research.

3. The results presented are sufficient to validate the theoretical findings and showcase the effectiveness of the proposed approach.

**Weaknesses:**

1. While the paper shows promising results, it is still outperformed by standard diffusion models, especially in terms of FID scores. Further work might be necessary to reach SOTA performance on larger datasets like ImageNet.

2. Related to the previous point, the experimental results are primarily limited to smaller datasets (CIFAR10, MNIST, CELEB-A, and LSUN Church), which may not reflect the potential scalability of DMMD to more complex, high-resolution datasets.

3. Although the method avoids adversarial training, the noise-adaptive MMD flow still introduce complexity, which may limit reproducibility.

**Questions:**

1. As mentioned above, what challenges do you consider in scaling DMMD to larger datasets like ImageNet?
2. What factors limit DMMD's FID performance relative to state-of-the-art diffusion models?

---

> ### Author Response · Authors · 2024-11-21
> **Answer to Reviewer QpwN**
>
> We thank the reviewer QpwN for the positive feedback on our work. Please find our answers below.
>
> > While the paper shows promising results, it is still outperformed by standard diffusion models, especially in terms of FID scores. Further work might be necessary to reach SOTA performance on larger datasets like ImageNet.
>
> Please see our common answer above.
>
> > Related to the previous point, the experimental results are primarily limited to smaller datasets (CIFAR10, MNIST, CELEB-A, and LSUN Church), which may not reflect the potential scalability of DMMD to more complex, high-resolution datasets.
>
> Please see our common answer above.
>
> > Although the method avoids adversarial training, the noise-adaptive MMD flow still introduce complexity, which may limit reproducibility.
>
> Sampling from the noise-adaptive MMD flow introduces complexity in the sense that one must choose the hyperparameters for the sampling process. The problem of hyperparameter choice at sampling time also arises for diffusion models – i.e., the performance of diffusion models differs drastically depending on the sampling scheme chosen. Details on hyperparameter choice are provided in Appendix D.
>
> > As mentioned above, what challenges do you consider in scaling DMMD to larger datasets like ImageNet?
>
> See above: the implementation for high resolution images like imagenet would require defining an MMD diffusion in latent space. This is an important topic for future work.
>
> > What factors limit DMMD's FID performance relative to state-of-the-art diffusion models?
>
> Please see above for our common answer.

---

> > ### Comment · Reviewer_QpwN · 2024-11-27
> >
> > Dear Authors,
> >
> > Thank you for providing further clarification regarding FID scores and extending your results to a dataset with a size of 128x128. I believe further improvement in these aspects will enhance the quality and impact of the paper. At this point, I have no further questions and will keep my score the same.

---

### Author Response · Authors · 2024-11-21
**Joint answer**

Please find below a common answer to the occurring points raised during the review process.

# DMMD does not achieve SOTA on Image Generation datasets

Please note that the focus of our work was to demonstrate that it is possible to learn a Wasserstein gradient flow based on a GAN-like discriminator, without adversarial training, and to use it to produce samples with this flow. As it stands now, our method delivers the best empirical results in the class of methods which rely on Wasserstein gradient flows (see our related work and experiments section), and are the first such results to come close to score-based diffusion approaches.

As noted by reviewers, DMMD nonetheless still underperforms compared to the standard diffusion models and these limitations will be the focus for the future work.

State of the art diffusion models have benefited from an intensive search effort over parameters and architectures by the research community, focusing on optimizing their performance on datasets like CIFAR-10. In fact, it took the community 5 years to improve the originally proposed diffusion model [1] to achieve good performance (see DDPM [2]). We believe pushing DMMD’s FID to be closer to diffusion models would require to re-evaluate the design choices made in DMMD (such as architecture, forward noising process, etc) and carefully tune these to the best possible performance on CIFAR-10. We presently borrow settings and architectures  which may not be optimal for DMMD, yet the performance of DMMD is already better than all existing methods employing Wasserstein gradient flows. Further work on tuning parameters and architectures is an important direction for future work.

# Scaling DMMD to high resolution datasets

In our experiments, we considered CIFAR-10 (50k samples) and MNIST (60k samples) to be 32x32 datasets and CELEB-A (162k samples) and LSUN Church (126k samples) to be 64x64 datasets. In principle, scaling to larger image sizes will entail the same increase in complexity as for diffusion models. In fact, in the literature it is common [3-4] to use latent diffusion models for large image sizes, effectively reducing the size of the images to a much smaller, latent dimensionality. The implementation of the Diffusion MMD in latent space for larger image sizes is an interesting topic for future work.

For the purpose of this rebuttal, we scaled DMMD to CELEB-A-HQ (30k examples) of size 128x128. We used the same backbone as in the DDPM paper [4] for the CELEB-A-HQ-256x256 dataset.

We train both DDPM and DMMD on these datasets. Both methods use unconditional sampling with 100 NFEs. For DMMD at sampling time, we use $T=10$ noise levels with $N_{s}=10$ and $\eta=0.001$ gradient flow learning rate and with $N_p=50$ noisy particles. See Alg. 2 for details.

Our results:

* DDPM -- FID=11.67, Inception Score (IS)=2.87
* DMMD -- FID=14.09, IS=2.57

We provide samples in the anonymized link  https://tinyurl.com/iclr2025images.

The results suggest that DMMD still have a performance gap compared to DDPM as observed for lower resolution datasets. Training on yet higher resolution images (256x256) currently presents memory challenges for our training hardware, which we were not able to overcome in time for the rebuttal. We are working to obtain higher resolution results, but we believe that the 128x128 results are already representative of expected performance.

We believe that future improvements in parameter and architecture refinement for DMMD (see our point about SOTA) will translate into improved performance of DMMD in these high resolution datasets. This is left for future work.

We will add these results to the camera-ready version of our paper.

# Computational complexity

We add more details about the computational complexity of DMMD during training and sampling time. We will add this information in the Appendix.

## Training-time complexity

Computing total loss in training iteration (see Alg. 1) on a batch of B samples is $O(B^2)$for an arbitrary kernel and $O(B)$ for a linear kernel (see Appendix B.3). We compute it for $N_n$ noise levels meaning that the cost scales as $O(N_nB^2)$ for an arbitrary kernel and $O(N_nB)$ for a linear kernel. This is more expensive than DDPM [2], which scales as $O(B)$.

## Sampling-time complexity

During sampling time (see Alg. 2), we use $T$ noise levels and do $N_s$ steps per noise level with $N_p$ noisy particles. Let $O(F)$ be the cost of forward pass for the backbone network. The gradient takes $O(2F)$ time to compute. Each sampling step for DDPM costs $O(F)$, whereas for DMMD it costs $O(FN_p^2)$ for an arbitrary kernel, $O(FN_p)$ for linear kernel and $O(F)$ when using approximate sampling procedure.

**References**:

[1] Deep Unsupervised Learning using Nonequilibrium Thermodynamics, Jascha Sohl-Dickstein, Eric A. Weiss, Niru Maheswaranathan, Surya Ganguli

[2] Denoising Diffusion Probabilistic Models, Jonathan Ho, Ajay Jain, Pieter Abbeel

[3] Imagen 3

[4] Dalle-2

---

### Meta-Review · Area_Chair_HSZT · 2024-12-22

**Metareview:**

The paper presents an approach to generative modeling by combining Maximum Mean Discrepancy (MMD) with diffusion models, avoiding the complexities of adversarial training. Reviewers appreciated the paper’s mathematical rigor, particularly its noise-adaptive gradient flow mechanism and efficient approximation of sampling using linear kernels. The framework offers valuable theoretical insights, and its empirical validation on datasets such as CIFAR-10, MNIST, CelebA, and LSUN Church showcases competitive results against some baseline methods. These strengths underline the potential of DMMD in addressing challenges in adversarial training. However, reviewers expressed concerns regarding the framework’s scalability and performance relative to state-of-the-art diffusion models. The DMMD framework has yet to demonstrate comparable results on larger or high-resolution datasets, such as ImageNet, and its computational cost during the generation process remains significant, especially when using non-linear kernels. Experiments primarily focused on lower-resolution datasets, limiting the evidence of scalability to more complex datasets. Additionally, the framework requires access to the training dataset for certain kernels, which might hinder general applicability. These issues highlight the need for further work to validate its robustness and efficiency on high-dimensional data. After the rebuttal phase the work remains borderline. With moderate revisions and a focus on addressing scalability and methodological concerns, the DMMD paper has the potential to make a significant contribution to generative modeling research.

**Additional Comments On Reviewer Discussion:**

- The rebuttal led to an increase in scores, however the work remains borderline.
- The authors raised an issue with reviewer xQpY and pointed out that they copied their exact response from a previous conference (evidence is there). I would suggest removing the reviewer from the pool. I have considered this when making the final decision.

---

### Decision · Program_Chairs · 2025-01-22

Accept (Poster)